# Life-cycle-coupled evolution of mitosis in close relatives of animals

Hiral Shah[1 ✉], Marine Olivetta[2], Chandni Bhickta[1,6], Paolo Ronchi[3,6], Monika Trupinić[4,6], Eelco C. Tromer[5], Iva M. Tolić[4], Yannick Schwab[1,3], Omaya Dudin[2 ✉] & Gautam Dey[1 ✉]

Eukaryotes have evolved towards one of two extremes along a spectrum of strategies for remodelling the nuclear envelope during cell division: disassembling the nuclear envelope in an open mitosis or constructing an intranuclear spindle in a closed mitosis[1,2]. Both classes of mitotic remodelling involve key differences in the core division machinery but the evolutionary reasons for adopting a specific mechanism are unclear. Here we use an integrated comparative genomics and ultrastructural imaging approach to investigate mitotic strategies in Ichthyosporea, close relatives of animals and fungi. We show that species in this clade have diverged towards either a fungal-like closed mitosis or an animal-like open mitosis, probably to support distinct multinucleated or uninucleated states. Our results indicate that multinucleated life cycles favour the evolution of closed mitosis.

Eukaryotic mitosis relies on a tight coordination between chromosome segregation and the remodelling of the nuclear compartment to ensure the fidelity of nuclear division and genome inheritance[2,3]. Two classes of nuclear remodelling have been widely investigated: open mitosis[4], in which the nuclear envelope (NE) is disassembled at mitotic entry and reassembled following chromosome segregation; and closed mitosis[5–7], in which the nuclear compartment retains its identity throughout division (Extended Data Fig. 1). Although open and closed mitosis have each probably evolved independently several times in different branches of the eukaryotic tree[8,9], with many unique lineage-specific adaptations resulting in a broad distribution of intermediates from fully open to fully closed[1,2], the evolutionary pressures which drive species towards the extremes of either mitotic strategy are not well understood. Studies, primarily in mammalian and yeast models, suggest that open and closed mitosis require distinct adaptations in key structural components of the division machinery[1], including the microtubule organizing centre (MTOC)[10], the spindle[11], the NE[12,13] and the kinetochore[14]. For example, building an intranuclear spindle in closed mitosis must be accompanied by NE fenestration to allow insertion of the MTOC[15]. On the other hand, open mitosis requires distinct interphase and postmitotic mechanisms for the insertion of new nuclear pore complexes (NPCs) into the NE[16]. These significant differences in the core division machinery imply that certain molecular signatures of the mode of mitosis may be encoded in the genome, enabling the use of comparative genomics to identify new cases of probable divergence between related species outside traditional model systems. We can then combine phylogenetic inference with the targeted experimental investigation of mitotic dynamics in these lineages to ask whether constraints imposed by ecological niche and life cycle could drive species towards either open or closed mitosis.

The Opisthokonta, a principal eukaryotic group which includes animals, fungi and their deep-branching relatives, with the Amoebozoa as a close outgroup, present an ideal context for such an evolutionary

cell biology analysis, with species in the clade exhibiting a broad range of genome organization modes, physiology and ecology[17–19]. Importantly, either open or closed mitosis is dominant in the main animal and fungal lineages, respectively[2,3]. We know little about mitosis in the deep-branching opisthokonts which lie between animals and fungi, including the Choanoflagellatea, Filasterea, Ichthyosporea and Corallochytrea (Fig. 1a,b)[17,20]. Among these, Ichthyosporea, consisting of two main lineages, Dermocystida and Ichthyophonida, exhibit diverse life cycles (Fig. 1a) featuring a mixture of fungal-like traits and transient multicellular stages reminiscent of early animal development[17,21,22]. Most Ichthyosporea proliferate as coenocytes, multinucleated cells formed through sequential rounds of mitosis without cytokinesis, which eventually complete their life cycle through coordinated cellularization[21]. However, a few understudied species undergo nuclear division with coupled cell cleavages (palintomic division)[23,24], providing a unique opportunity to assess if and how mitotic strategies in a group of related species might be coupled to distinct uninucleated or multinucleated life cycles (Fig. 1a).

We first analysed the phylogenetic distributions of a set of conserved protein families (Source Data Fig. 1) involved in structural changes of the NE, chromosomes and spindle during mitosis (Fig. 1b and Extended Data Fig. 1). We identified putative orthologues and inferred gene trees of these key regulators across a set of representative opisthokonts, centred on the two main lineages of Ichthyosporea, the Ichthyophonida and Dermocystida, and using the Amoebozoa as a neighbouring outgroup which includes the model slime moulds *Dictyostelium* and *Physarum*[25] (Fig. 1b and Extended Data Fig. 1). Our results highlight features shared by all Ichthyosporea, such as the kinetochore-localized Dam complex also present in most fungi[26], as well as features restricted to specific ichthyosporean lineages. In an example of the latter, the ichthyophonid *Sphaeroforma arctica* (Fig. 1b) lacks centriolar components such as Sas6 and Plk4. *S. arctica* lacks many parts of the constitutive

[1]Cell Biology and Biophysics, European Molecular Biology Laboratory (EMBL), Heidelberg, Germany. [2]Swiss Institute for Experimental Cancer Research, School of Life Sciences, Swiss Federal Institute of Technology (EPFL), Lausanne, Switzerland. [3]Electron Microscopy Core Facility, European Molecular Biology Laboratory, Heidelberg, Germany. [4]Division of Molecular Biology, Ruđer Bošković Institute (RBI), Zagreb, Croatia. [5]Cell Biochemistry, Groningen Biomolecular Sciences & Biotechnology Institute, University of Groningen, Groningen, The Netherlands. [6]These authors contributed equally: Chandni Bhickta, Paolo Ronchi, Monika Trupinić. ✉e-mail: hiral.shah@embl.de; omaya.dudin@epfl.ch; gautam.dey@embl.de

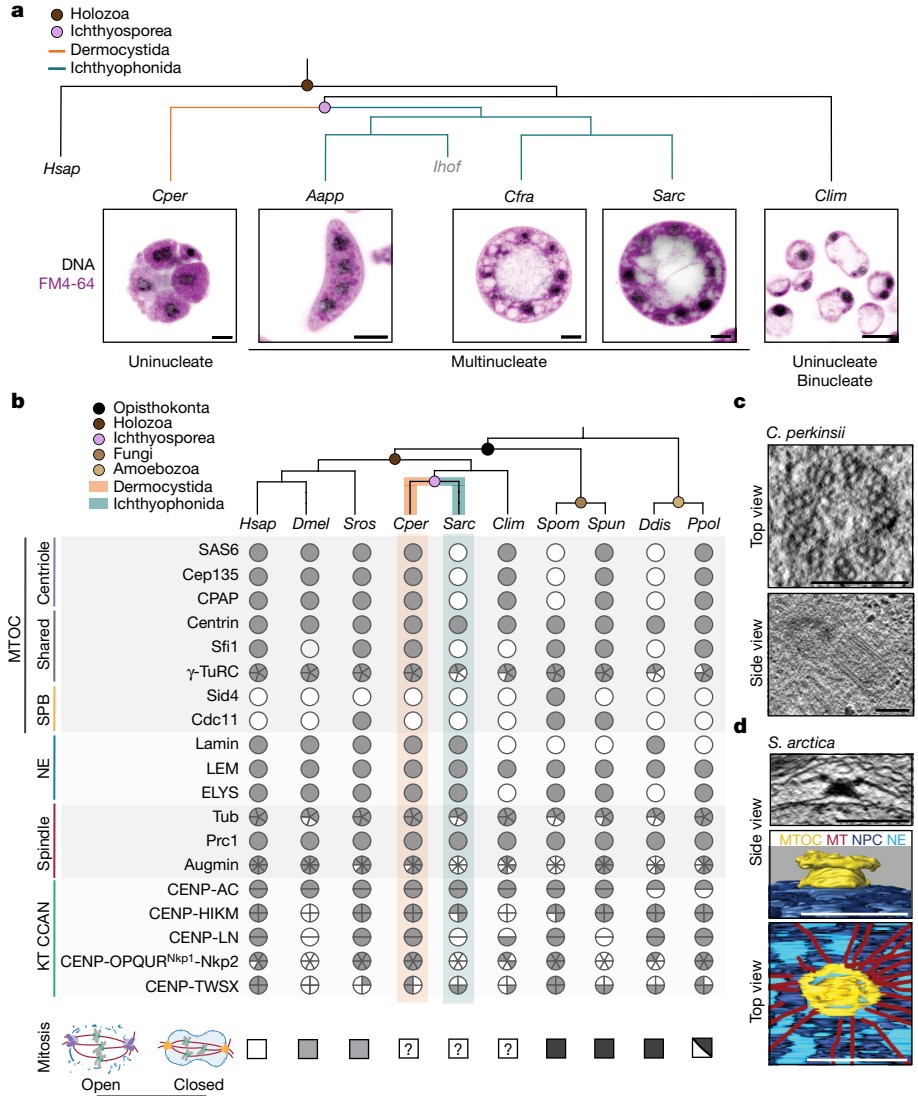

**Fig. 1 | Divergence of mitotic machinery in the Ichthyosporea with different life cycles. a**, Differences in life cycles and the uninucleated and multinucleated states of dermocystid *C. perkinsii* (*Cper*), ichthyophonids *A. appalachense* (*Aapp*), *S. arctica* (*Sarc*), *C. fragrantissima* (*Cfra*) and corallochytrean *C. limacisporum* (*Clim*), respectively. *Ihof, Ichthyophonus hofleri*. Representative image single-slice images through cells labelled for cell membranes with FM4-64 (magenta) and DNA (grey). **b**, Cladogram of opisthokonts, highlighting the position of Ichthyosporea between well-studied animal, fungal and amoebozoan model systems. Phylogenetic profiles of selected proteins involved in mitosis (complete profiles in Extended Data Fig. 1). Filled and empty circles or pie charts indicate the presence and absence of proteins, respectively (Methods). In addition to Ichthyosporea (Sarc and Cper) and Corallochytrea (Clim), profiles of key species are represented, including

*Homo sapiens* (Hsap), *Drosophila melanogaster* (Dmel), *S. pombe* (Spom), the choanoflagellate *Salpingoeca rosetta* (Sros), the early-branching chytrid fungus *Spizellomyces punctatus* (Spun) and amoebozoa *Dictyostelium discoideum* (Ddis) and *P. polycephalum* (Ppol). The mitotic strategies, open (white), intermediate (grey squares) or closed (dark grey squares), of the represented opisthokont and amoebozoan species are indicated at the end of each profile, KT (kinetochore). **c**, *C. perkinsii* has a centriolar MTOC. Single slices from TEM tomography of *C. perkinsii* cells showing top and side views through centrioles. **d**, *S. arctica* has an acentriolar MTOC. Single slice from TEM tomography of *S. arctica* interphase nucleus. Side and top views of segmentation of *S. arctica* MTOC from an interphase nucleus. Scale bars, 2 μm (**a**), 200 nm (**c**), 500 nm (**d**).

centromere-associated network (CCAN), which (although widely present across eukaryotes) has been lost in several lineages, including the Diptera (Fig. 1b and Extended Data Fig. 1). By contrast, the dermocystid *Chromosphaera perkinsii* has an animal-like repertoire of mitosis-related components, including a centriolar MTOC and the CCAN (Fig. 1b and Extended Data Fig. 1). Examining the proteomes of *S. arctica* and *C. perkinsii* in greater depth suggests more differences in spindle morphology and NPC components between the two groups. For example, *S. arctica* seems to be missing all subunits of the augmin complex responsible for nucleating new microtubules on existing spindle microtubule bundles[27]. Although all Ichthyosporea seem to

possess a PRC1/Ase1 spindle crosslinker[28,29], the gene tree suggests that the *S. arctica* PRC1 is positioned on one side of an ancient duplication, clustering away from typical animal and fungal PRC1, whereas *C. perkinsii* retains both orthologues (Extended Data Fig. 2a). The conserved nucleoporin ELYS is a key regulator of postmitotic NPC assembly, thought to be dispensable for pathways of interphase assembly and in systems with a closed mitosis. *C. perkinsii* possesses a full-length ELYS orthologue, whereas the truncated *S. arctica* ELYS is reminiscent of the fission yeast protein (Extended Data Fig. 2b,c). Although collectively these differences are a strong indication of a divergence in mitotic mode between *C. perkinsii* and *S. arctica*, they are not sufficient to determine

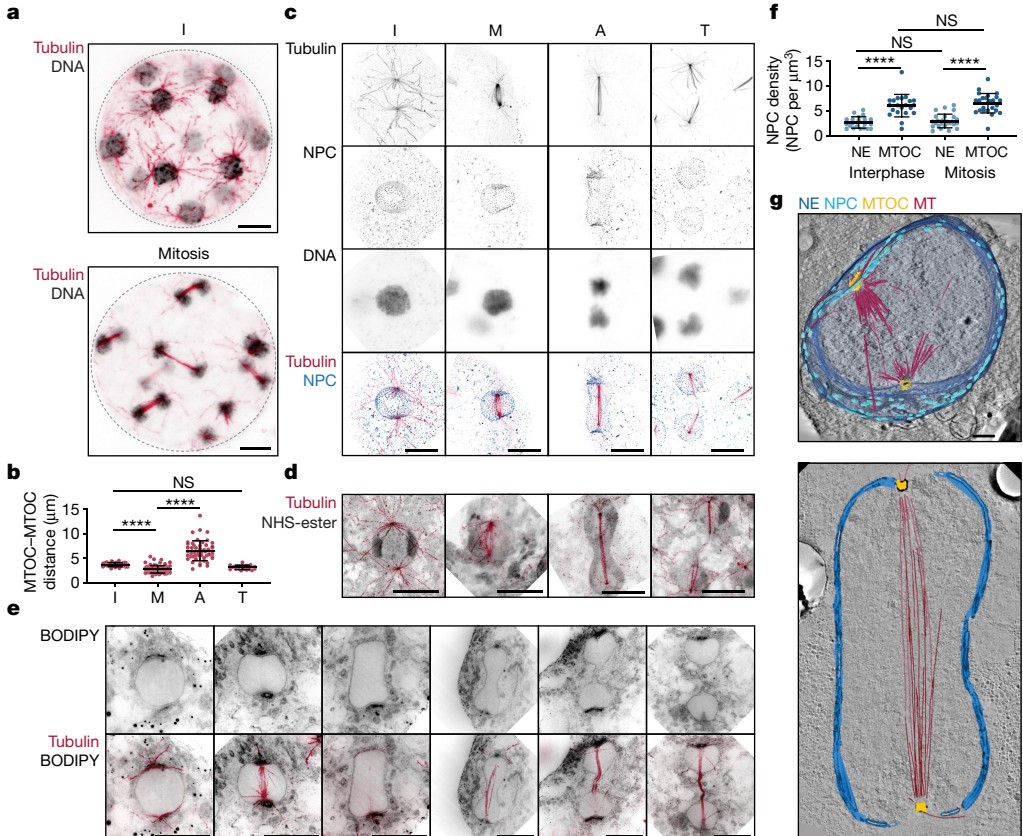

**Fig. 2 | *S. arctica* undergoes closed mitosis. a**, Representative maximum intensity projections of synchronized *S. arctica* interphase (top) and mitotic (bottom) coenocytes labelled for tubulin (red) and DNA (grey). **b**, Dot plot showing MTOC–MTOC distances for interphase (I), metaphase (M), anaphase (A) and telophase (T) ($n_I = 29$, $n_M = 75$, $n_A = 49$, $n_T = 19$) nuclei. Three biological replicates. In telophase, distance was measured from the MTOC to the end of spindle microtubules (MTs). Statistical analysis using a Kruskal–Wallis test; $P < 0.0001$ with Dunn's test for several comparisons. Adjusted $P$ values, I versus M; $P < 0.0001$, M versus A; $P < 0.0001$, I versus T; $P = 0.1909$. Data are mean ± s.d. **c**, Mitotic microtubule (red), DNA (grey) and NPC (blue) configurations. Representative maximum intensity projections from three biological replicates. **d**, *S. arctica* carries out closed mitosis forming a dumbbell-shaped nucleus in anaphase. Representative maximum intensity projections of nuclei with pan labelling (NHS-ester, grey) and tubulin labelling (red) from three biological

replicates. **e**, NE is maintained during *S. arctica* mitosis. Representative maximum intensity projections of central slices of nuclei with membrane (BODIPY, grey) and tubulin labelling (red). **f**, NPC density is maintained throughout the life cycle. Dot plot showing NPC density in interphase and mitosis, either proximal to the MTOC or per nucleus ($n_{interphase} = 21$, $n_{mitosis} = 29$ nuclei) (\*\*\*\*$P < 0.0001$; NS, not significant). Statistical analysis using a two-tailed Mann–Whitney $U$-test; I-NE versus M-NE $P = 0.5266$; I-SPB versus M-SPB $P = 0.3403$; I-NE versus I-SPB and M-NE versus M-SPB $P < 0.0001$. From three biological replicates. Data are mean ± s.d. **g**, Single slice from a representative electron tomogram volume of early (top) and late dumbbell-shaped (bottom) *S. arctica* mitotic nuclei overlaid with a 3D model showing the intranuclear spindle (red), MTOC (yellow) and NE (blue) and NPCs (light blue). Scale bars, 5 μm (**a,c,d,e**), 500 nm (**g**).

the position of either species along the spectrum from open to closed. Furthermore, inspection of the phylogenetic profiles of the other Ichthyosporea and assorted Holozoa (the group containing animals and their closest relatives) (Extended Data Fig. 1) reveals a significant degree of compositional variability in the complement of mitosis-associated protein complexes, making it difficult to draw any generalizable conclusions in the absence of an accompanying experimental investigation.

## Distinct MTOCs in Ichthyosporea

Therefore, we first set out to examine the MTOCs of *C. perkinsii*, the only free-living dermocystid isolated until now, and *S. arctica*, the best-studied ichthyosporean model overall[21,30,31], using transmission electron microscopy (TEM) and focussed ion beam scanning electron microscopy (FIB-SEM). As predicted by the phylogenetic analysis, we identified centrioles in *C. perkinsii* with canonical ninefold symmetry and a diameter (220 ± 14 nm, $n = 17$ centrioles) (Fig. 1c) very close to that of the typical animal centriole[32]. By contrast, we find that the

*S. arctica* MTOC is a multilayered structure positioned at the outer nuclear membrane during interphase (Fig. 1d and Supplementary Video 1), reminiscent of the fungal spindle pole body (SPB) and consistent with previous reports[19,33]. *S. arctica* MTOCs are duplicated even in the presence of hydroxyurea, an S-phase inhibitor, suggesting that duplication either occurs early in G1/S or proceeds independently of S phase (Extended Data Fig. 3a,b). During mitosis, the localization of the MTOC shifts from the outer to the inner nuclear membrane (Extended Data Fig. 3c,d), predictive of an intranuclear spindle. A single nuclear pore was located directly underneath a subset of interphase MTOCs (Extended Data Fig. 3d and Supplementary Video 1), possibly an intermediate in an insertion–extrusion cycle of the type best characterized in the fission yeast *Schizosaccharomyces pombe*[15]. The presence of an animal-like centriole-based MTOC in *C. perkinsii* (Fig. 1b, c and Extended Data Fig. 1) and a unique NE-associated acentriolar MTOC in *S. arctica* with some fungal features (Fig. 1b,d and Extended Data Fig. 3), validated our combined phylogenetic and comparative cell biology approach and enabled us to focus on the accompanying mitotic strategies in the two species.

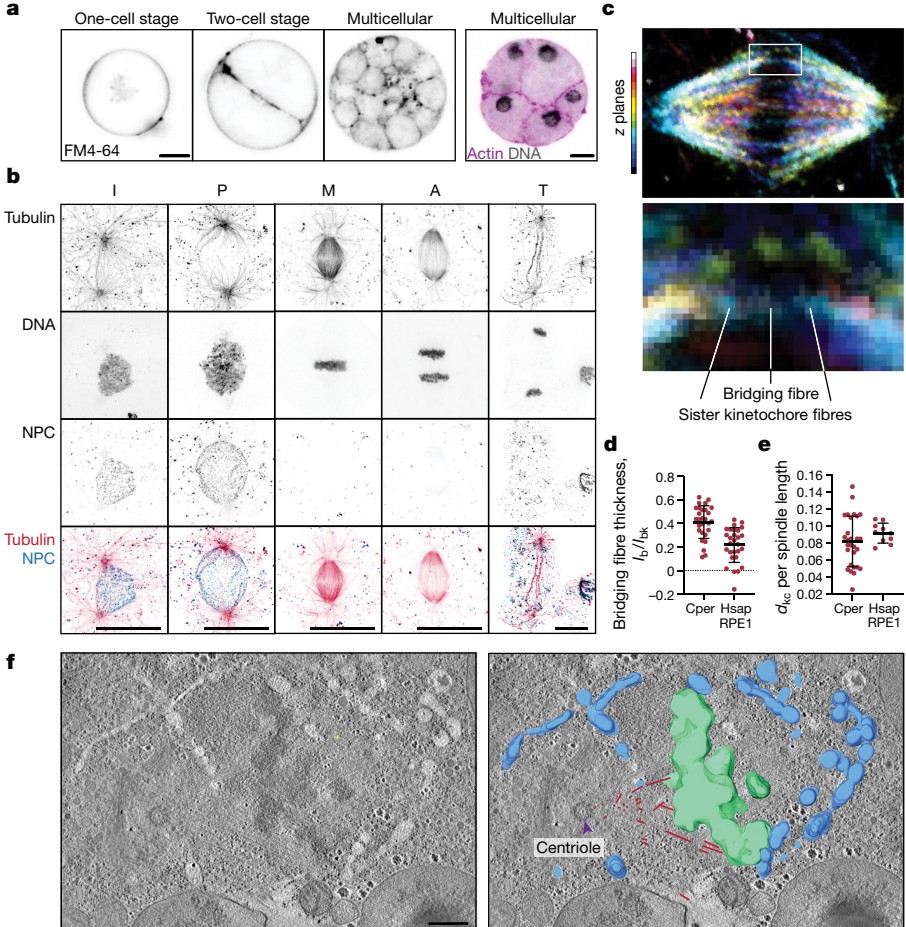

**Fig. 3 | *C. perkinsii* undergoes open mitosis. a**, Representative images of cell cleavages in *C. perkinsii* cells at different life cycle stages labelled for: left, cell membrane with FM4-64; right, actin (purple) and DNA (grey). **b**, Representative maximum intensity projections of *C. perkinsii* nuclei at different stages of the cell cycle (I, P, M, A and T), in cells labelled for tubulin (red), DNA (grey) and NPCs (blue). **c**, *C. perkinsii* spindle in metaphase coloured for depth as shown in the legend on the left (top); enlarged section of the spindle showing kinetochore and bridging fibres (bottom) analysed in **d** and **e** and Extended Data Fig. 8. **d**, Bridging fibre thickness relative to the kinetochore fibre thickness

($I$, intensity); $P < 0.0001$. **e**, Interkinetochore distance ($d$) scaled to the spindle long axis; $P = 0.1512$. Data for RPE1 cells in **d** and **e** were taken from ref. 27 $n_{Cperk} = 28$, $n_{RPE1} = 10$ spindles. Spindles were analysed over three independent experiments. We performed statistical analysis using two-tailed Student's *t*-test. Data are mean ± s.d. **f**, Single slice from a representative electron tomogram of *C. perkinsii* mitotic nucleus overlaid with a 3D model showing the spindle microtubules (red), chromosomes (green) and NE (blue). The purple arrowhead indicates the centriole in focus. Scale bars, 5 μm (**a**,**b**), 500 nm (**f**). P, prophase.

## Fungal-like closed mitosis in *S. arctica*

*S. arctica* proliferates through synchronized rounds of nuclear divisions without cytokinesis, resulting in the formation of multinucleated coenocytes which later undergo actomyosin-dependent cellularization driven by the nuclear-to-cytoplasm ratio[21,31]. *S. arctica* live cells exclude the dye FM4-64 from the nuclear volume[31], including during mitosis. This enabled us to observe a reproducible sequence of mitotic shape changes in *S. arctica* nuclei, including the formation of a dumbbell late in the division process, without any detectable change in the integrity of the NE barrier (Extended Data Fig. 4 and Supplementary Video 2). These features are indicative of a fungal-like closed mitosis, prompting a detailed investigation of the architecture of the MTOC, microtubule cytoskeleton, NE and NPCs using TEM, FIB-SEM and immunofluorescence in synchronized *S. arctica* cells. Classical immunofluorescence protocols are ineffective in Ichthyosporea, primarily because of the presence of a thick cell wall of unknown composition (36 ± 19% of cells stained in typical experiments, with many cells deforming following permeabilization; Extended Data Fig. 5a). To overcome this challenge, we implemented ultrastructure expansion microscopy (U-ExM), dramatically improving both the

staining efficiency (92 ± 6% of cells stained; Extended Data Fig. 5a) and the spatial resolution of immunofluorescence images through fourfold isotropic expansion (Extended Data Fig. 5b). U-ExM revealed a prominent polar MTOC-nucleated astral microtubule network in both interphase and mitotic *S. arctica* cells (Fig. 2a,c–e). We reconstructed the spatiotemporal dynamics of *S. arctica* mitosis by ordering nuclei along a pseudo-timeline inferred from nuclear shapes as well as spindle length (Fig. 2b–e, Extended Data Fig. 4 and Supplementary Video 2). We observe intranuclear spindle halves initially coalescing to reduce the distance between the two MTOCs (Fig. 2b–e and Supplementary Videos 3 and 5). During anaphase, the spindle, now linear and bundled, elongates to separate both DNA masses with minimal apparent chromosome condensation (Fig. 2b,c and Supplementary Video 4). Using pan protein labelling and mAb414 which targets NPCs in combination with electron tomography, we found that anaphase nuclei take on the characteristic dumbbell shape commonly observed in amoebozoan and fungal closed mitosis (Fig. 2d,g and Supplementary Videos 4 and 6). Using the membrane marker BODIPY ceramide, we confirmed that the NE is preserved throughout mitosis with NPCs maintaining their localization patterns and density at a constant level, both defining features of a closed mitosis (Fig. 2e,f). Treatment with

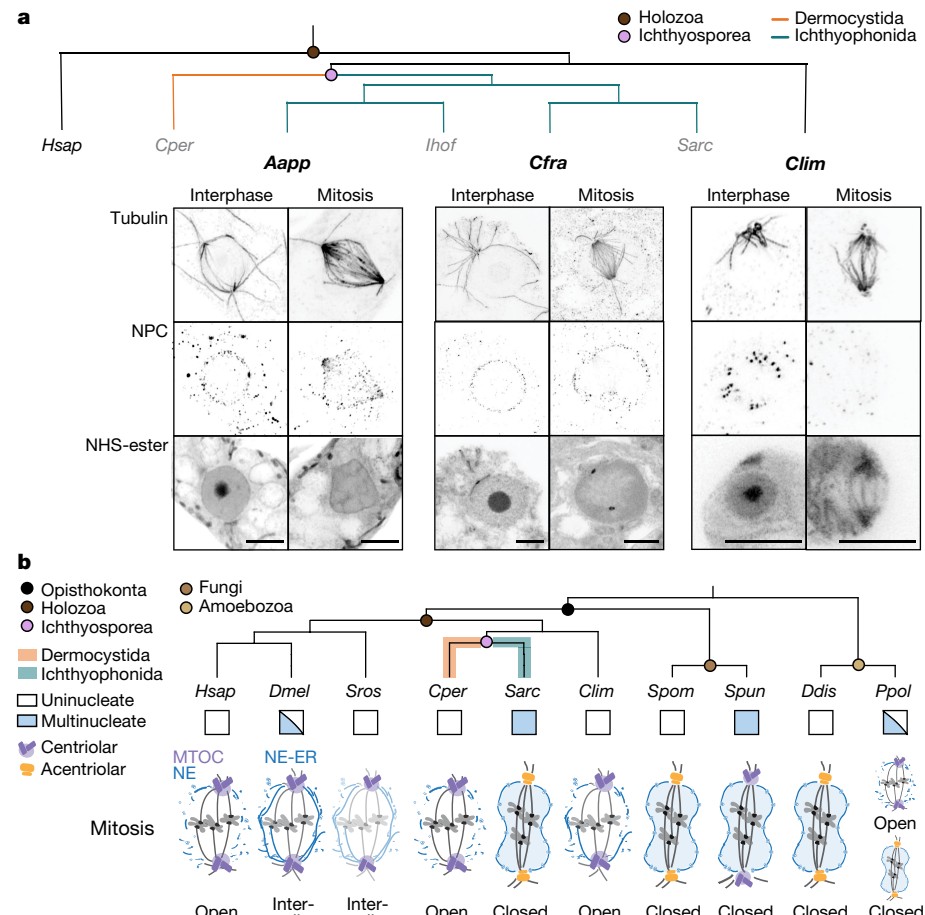

**Fig. 4 | Specialization of mitotic strategies coupled to distinct life cycles in opisthokonts. a**, Representative maximum intensity projections of central slices of *A. appalachense, C. frangrantissima* and *C. limacisporum* nuclei during interphase (left) and mitosis (right) labelled for tubulin, NPCs, DNA and NHS-ester. **b**, Ichthyophonida, including the model ichthyosporean, *S. arctica* (Sarc) form multinucleated cells, like many fungal and amoebozoan species (Ddis and Ppol), through a series of closed mitoses. As in classical fungal closed mitosis, ichthyosporean closed mitosis involves NE embedding of the acentriolar MTOC and remodelling of the nucleus into dumbbells on elongation of the intranuclear spindle. Dermocystids, on the contrary, divide by cell cleavages resulting in uninucleated cells and undergo centriole-mediated open mitosis with NE breakdown involving spindle architecture reminiscent of mammalian (Hsap) mitosis. The corallochytrean *C. limacisporum*, which divides as uninucleated cells through open mitosis, suggests that open mitosis is ancestral in Holozoa. In coenocytic insect embryos (Dmel), the integrity of the nuclear compartment is maintained by the ER (endoplasmic reticulum). Across opisthokonts and amoebozoan (Ppol) outgroups, the data suggest a coupling between open or closed mitosis with uninucleated and multinucleated life cycles. Scale bars, 2 μm (**a**).

cerulenin, a drug which blocks the lipid synthesis required for NE expansion in closed mitosis[34] but has little impact on open mitosis systems, arrested cells early in their life cycle. These cells contained large multipolar nuclei resulting from a failed nuclear division, producing in turn aberrant multipolar spindles and mitotic failure (Extended Data Fig. 6a,b). Remarkably, we observed a radial arrangement of the NPCs surrounding the MTOCs (Extended Data Fig. 7a–c). These radial arrays, also present in other *Sphaeroforma* species (Extended Data Fig. 7d), align along cytoplasmic microtubules emerging from the MTOCs and reorganize on microtubule depolymerization using carbendazim (MBC) (Extended Data Fig. 7e). Mild MBC treatment causes the central spindle to collapse, although without perturbing NE integrity (Extended Data Fig. 6c), and a few NE-adjacent astral microtubules persist (Extended Data Fig. 6c–f), indicating that the putative microtubule–NPC interaction at the NE might enhance microtubule stability. Finally, we confirmed using TEM tomography and FIB-SEM that the NE remains intact throughout mitosis (Fig. 2g and Supplementary Videos 5 and 6). Together, phylogenetic analysis supported by ultrastructure imaging and functional inhibition using small molecules demonstrates that the coenocytic life cycle of *S. arctica* is accompanied by closed mitosis mediated by a unique acentriolar MTOC reminiscent of fungi.

## Animal-like open mitosis in *C. perkinsii*

We then investigated mitosis in *C. perkinsii*, the most animal-like ichthyosporean based on its phylogenetic profile (Fig. 1b and Extended Data Fig. 1). Similar to its close parasitic relative *Sphaerothecum destruens*[23], *C. perkinsii* proliferates through palintomic divisions or cell cleavages, thus exhibiting a uninucleate life cycle (Fig. 3a). Reconstructing the sequence of *C. perkinsii* mitotic stages reveals a strikingly human-like spindle nucleated from centrioles (Fig. 3b and Supplementary Videos 7 and 8), with several key structural similarities, including the presence of kinetochore fibres and bridging fibres, scaling of interkinetochore distances and twist of the spindle (Fig. 3c–e, Extended Data Fig. 8 and Supplementary Information), an equatorial arrangement of condensed chromosomes in metaphase (Fig. 3b) and the simultaneous segregation of chromosome complements to opposing poles in anaphase (Fig. 3b). By contrast to *S. arctica* nuclear division, *C. perkinsii* mitosis is characterized by NE breakdown and reassembly kinetics typical of

mitosis in human cells (Fig. 3b,f and Supplementary Videos 7–9), with NPCs disappearing in prophase accompanied by a loss of NE integrity (Fig. 3b,c) and reappearing in telophase (Fig. 3b).

To provide a broader context for the sharp divergence in mitotic mode between dermocystids, exemplified by *C. perkinsii* and ichthyophonids, typified by *S. arctica*, we examined next a range of further ichthyosporean species and their closest living outgroup, the corallochytrean *Corallochytrium limacisporum*[35]. Using U-ExM of microtubules, NE and NPCs, we find that other coenocytic ichthyophonids (three more *Sphaeroforma* species as well as *Creolimax fragrantissima* and *Amoebidium appalachense*), seem to undergo a closed mitosis with the persistence of NPCs and an intact nuclear boundary as inferred from pan labelling (Fig. 4a and Extended Data Fig. 9). U-ExM and TEM tomography of *Amoebidium* cells uncovered cytoplasmic as well as nucleus-associated acentriolar MTOCs (Extended Data Fig. 9d) and, somewhat surprisingly, given the phylogenetic data (Extended Data Fig. 1), we were unable to identify any centrioles either in mitotic or interphase cells. By contrast, and consistent with the phylogenetic analysis (Fig. 1a and Extended Data Fig. 1), the predominantly uninucleate life cycle of the outgroup corallochytrean *C. limacisporum*[36] seems to be facilitated by open mitosis which relies on a spindle nucleated from centriolar MTOCs (Fig. 4a and Extended Data Fig. 9c,e). Although *C. limacisporum* proliferates through binary fission, with a characteristic long gap between mitosis and cytokinesis which results in a large proportion of binucleated cells at steady state (Fig. 1a and Extended Data Fig. 10), a small proportion of cells have been reported to exist as coenocytes[36]. These coenocytes, representing at most 3% of the population, divide rarely and often asynchronously, reaching a maximum of eight irregularly spaced nuclei (Extended Data Fig. 10a–c)[36]. Although we observed no difference between the open mitosis of the uninucleated cells and that of the coenocytes, a limited capacity for coenocytic division in this species might be supported by asynchronous mitoses and partial remnants of the NE barrier observable by electron tomography (Extended Data Fig. 10e,f). By contrast, *C. perkinsii* cells forced to enter a multinucleated state through the induction of cytokinetic failure, exhibit multipolar spindles and major mitotic defects (Extended Data Fig. 10g–i), much like animal cells[37], highlighting the incompatibility of fully open mitosis with coenocytic divisions.

With most studies restricted to a handful of animal and fungal model systems, it has been challenging to provide a mechanistic and evolutionary basis for the extensive phenotypic diversity of mitotic modes across eukaryotes[1]. Here, we provide evidence for a striking life cycle-coupled divergence between open and closed mitosis in the Ichthyosporea (Fig. 4b), cementing their role as a key group of species, along with other deep-branching Holozoa, for investigating the evolution of mitosis. The centriole-dependent open mitosis of *C. limacisporum* (Fig. 4 and Extended Data Figs. 9 and 10) together with its limited capacity for coenocyte formation suggests several plausible scenarios for mitosis in the holozoan ancestor, including open, closed or an intermediate form, possibly accompanied by a complex life cycle which alternated between these states depending on whether it was multinucleated or uninucleate. Such a life cycle would be closely analogous to that of *Physarum polycephalum* (Fig. 4), which transitions from a crawling uninucleate amoeba with centrioles and open mitosis to a giant acentriolar coenocyte carrying out a closed mitosis[38]. From this putative ancestor, the dermocystids inherited a flagellated life cycle stage[24,35] and, as we show in this study for *C. perkinsii*, specialized towards palintomic divisions through open mitosis (Fig. 3 and Extended Data Fig. 8); features shared with the ancestor of animals. By contrast, the adoption of an exclusively coenocytic life cycle in the Ichthyosporea was accompanied by the loss of the centriole, the de novo evolution or retention of a distinct, nucleus-associated MTOC and a fully closed mitosis. The ichthyophonid pathway to specialization provides an intriguing parallel to events in fungal evolution[18]. Zooming out, the ability of species to adopt divergent mitotic strategies over relatively short evolutionary timescales, or even in the same life cycle as in *Physarum* (Fig. 4), suggests an explanation for the limited predictive power of our phylogenetic profiles: cells are able to repurpose the same core, conserved machinery for different mitotic strategies[6].

The broad range of examples of closed mitosis with centrioles (Fig. 4)[39,40] and open mitosis without centrioles[41,42] outside the Opisthokonta argues that the presence of a flagellum or basal body does not per se constrain the mode of mitosis. Instead, our results indicate that having a coenocytic life cycle stage, in which more than two nuclei must divide and faithfully segregate in a shared cytoplasm, requires a closed or semi-closed mitosis (Fig. 4 and Extended Data Fig. 10). This model could explain the semi-closed or closed mitosis observed in the *Drosophila* coenocytic embryo[43,44], the germline of various animal lineages[45,46] and hyphal fungi[7,47] and is probably broadly generalizable to other eukaryotes outside the Opisthokonta, as in apicomplexan parasites or the coenocyte of *P. polycephalum*[38,39]. A corollary of our hypothesis is that closed mitosis can persist even when the organism evolves a unicellular, uninucleate life cycle[6,48], as in yeasts evolving from hyphal fungal ancestors, but in such cases is apparently no longer under strict selection to remain closed[49].

Beyond mitosis, our work highlights that genotype alone, although a powerful hypothesis generator is insufficient to predict cellular phenotypes which are invariably constrained by ecological niche and life cycles. However, we have access to many more high-quality genomes than experimental model systems and developing a new species into a model system demands many years of dedicated effort. Here, we address that issue using a volumetric ultrastructure imaging approach which combines the scalable tools of expansion microscopy with the traditional advantages of electron microscopy. When integrated with phylogenetics, such a framework can enable a comparative approach to investigating diversity and evolution in cell biology.

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

## Methods

### Phylogenetic analysis

We generated profiles of representative proteins from mitosis-associated cellular components including the centrosome, SPB, spindle, NE and the kinetochore for early-branching animal and fungal lineages alongside model species with well-characterized mitosis. The proteomes of *H. sapiens*, *S. rosetta*, *Monosiga brevicollis*, *Capsaspora owczarzaki*, *S. pombe*, *S. punctatus*, *D. discoideum* and *P. polycephalum* were obtained from Eukprot v02.2020_06_30 (ref. 50). The ichthyosporean proteomes were obtained from recent phylogenomic studies[19,21,35]. To identify putative orthologues we first searched for homologues with human proteins using phmmer (HMMER 3.3.2, November 2020; http://hmmer.org/)[51]. In case of divergence or absence of human proteins, searches were also carried out with proteins from other model species including *S. pombe, D. melanogaster* and *D. discoideum*. This was followed by Hidden Markov model (HMM)-based searches with the associated PFAM (http://pfam.xfam.org)[52] models using hmmsearch (HMMER 3.3.2, November 2020)[51]. For proteins in which homologues were not recovered by existing HMMs, new HMMs were generated. The multiple sequence alignment was done with MAFFT v.7.490 using 'linsi' optimized for local homology[53]. The alignments were inspected and trimmed using TrimAl v.1.2 to remove the less-conserved regions[54]. The trimmed alignments were used for tree inference with IQTree v.2.0.3 2020 using the model finder and ultrafast bootstraps (1,000) bootstraps[55–57]. The trees were visualized and annotated using FigTree (http://tree.bio.ed.ac.uk/software/figtree/). This process was performed iteratively to obtain better alignments which gave trees with higher bootstrap values. The alignment was then used to generate an HMM using the hmmbuild command in HMMER. This process was performed iteratively while incorporating the newly discovered homologues in the next round. The protein sequences are provided as fasta files. For most kinetochore proteins, iterative similarity searches were performed using previously generated HMMs of which candidate genes were scrutinized on the basis of known domain and motif topologies[14]. Trees and phylogenetic profiles in Fig. 1 and Extended Data Fig. 1 were visualized using iTOL[58].

### Ichthyosporea cultures and growth conditions

The different Ichthyophonid species (*Sphaeroforma* sp.[31,59], *C. fragrantissima* and *A. appalachense*[35] were provided by the laboratory of O.D., whereas *C. limacisporum* and *C. perkinsii*[35] were kindly provided by H. Suga and all the species originate from the Multicellgenome laboratory in Barcelona. The model ichthyosporean *S. arctica* with established protocols for cell synchronization, live cell imaging and cytoskeletal inhibitor assays was selected as a representative Ichthyophonid for this study[21,30,31]. Our attempts at replicating the reported genetic transformation of *C. fragrantissima*[60] were unsuccessful. *S. arctica* cultures were maintained at 17 °C in marine broth (Difco, 37.4 g l⁻¹) and synchronized as previously described[21,30,31]. Briefly, for synchronization, 1/16 marine broth was prepared by diluting marine broth in artificial seawater (Instant Ocean, 37 g l⁻¹). Cultures were diluted 1:100 in 1/16 marine broth and grown for 3 days to obtain saturated cultures. To obtain a synchronized culture, the saturated cultures were inoculated 1:50 in fresh marine broth. To obtain the 8–32 nuclear stage, cells were fixed around 28.5 h after inoculation. Other *Sphaeroforma* sp., *S. gastrica*, *S. nootkatensis* and *S. napiecek* and *C. fragrantissima* were maintained at 17 °C in marine broth similar to *S. arctica*. *C. perkinsii, A. appalachense* and *C. limacisporum* were grown at 23 °C protected from light. For experiments with *C. perkinsii*, 6-day-old cultures were filtered using a 5 µm filter to obtain small newborn cells which are then diluted 1:100 in *C. perkinsii* medium to obtain synchronous cultures. The cells were fixed at the one to eight cell stage (60–90 h after dilution) to capture the initial mitotic events. *C. limacisporum* cultures were grown in marine broth. *A. appalachense* cultures were grown in Amoebidium medium (yeast extract 3 g, peptone 5 g, water 1 l, autoclaved and aseptically filtered). For maintenance, cultures were diluted 1:1,000 every 2 weeks and restarted from a cryopreserved stock every 6 months.

### Immunostaining

The cell culture flasks were scraped and the suspension was added to 15 ml Falcon flasks to sediment for 15–30 min. The supernatant was removed and cells were transferred to 1.5 ml microfuge tubes and fixative was added for 30 min. The cells were fixed with 4% formaldehyde in 250 mM sorbitol solution, washed twice with 1× phosphate buffer saline (PBS) and resuspended in 20–30 µl of PBS. Cells were permeabilized using nine freeze–thaw cycles (liquid N₂,10 s: 42 °C, 1 min). This was followed by blocking in 3% bovine serum albumin (BSA) in PBST (1× PBS with 0.1% Tween20). Primary antibody (Tubulin-E7 antibody DSHB, NB600-936 Novus Biologicals, AA344 and AA345 (ABCD antibodies and anti-NPC proteins−MAb414 Biolegend 902901)) was used at 1:500 to 1:1,000 and incubated at 4 °C overnight or 2–5 h at 37 °C. This was followed by three washes for 10 min at room temperature and addition of the secondary antibody. Goat anti-mouse secondary antibody, Alexa Fluor 488 (Thermo A-11001), Goat anti-guinea pig secondary antibody, Alexa Fluor 568 (Thermo A-11075), Goat anti-rabbit secondary antibody, Alexa Fluor 568 (Thermo A78955) were used as secondary antibodies at 1:500 to 1:1,000. Incubation was done at 4 °C overnight or 2–5 h at 37 °C. The cells were then washed and resuspended in fresh 1× PBS for imaging. DNA was stained with Hoechst 33352 at a final concentration of 0.4 µM. For live cell imaging, cells were stained with FM-464 at a final concentration of 10 µM.

### Ultrastructural expansion microscopy

U-ExM was performed as previously described[61]. Briefly, the cells were fixed with 4% formaldehyde in 250 mM sorbitol solution, washed twice with 1× PBS and resuspended in 20–30 µl of PBS. The fixed cells were then allowed to attach to 12 mm poly-ʟ-lysine-coated coverslips for 1 h. This was followed by anchoring in acrylamide/ formaldehyde (1% acrylamide/ 0.7% formaldehyde) solution for at least 5 h and up to 12 h at 37 °C. A monomer solution (19% (wt/wt) sodium acrylate (ChemCruz, AKSci 7446-81-3), 10% (wt/wt) acrylamide (Sigma-Aldrich A4058), 0.1% (wt/wt) *N*,*N*'-methylenebisacrylamide (Sigma-Aldrich M1533) in PBS) was used for gelation and gels were allowed to polymerize for 1 h at 37 °C in a moist chamber. For denaturation, gels were transferred to the denaturation buffer (50 mM Tris pH 9.0, 200 mM NaCl, 200 mM SDS, pH to 9.0) for 15 min at room temperature and then shifted to 95 °C for 1 h. Following denaturation, expansion was performed with several water exchanges as previously described[61]. After expansion, gel diameter was measured and used to determine the expansion factor. For all U-ExM images, scale bars indicate actual size; rescaled for gel expansion factor. Pan labelling of U-ExM was done at 1:500 with Dylight 405 (ThermoFischer, 46400) or Alexa Fluor NHS-Ester 594 (ThermoFischer, A20004) in 1× PBS or NaHCO₃ for 1.5 h or overnight. For membrane labelling, gels were stained with BODIPY TR ceramide (ThermoFischer D7540, 2 mM stock in dimethylsulfoxide) at 1:500 dilution in 1× PBS. Immunostaining was performed as mentioned above. All antibodies were prepared in 3% PBS with 0.1% Tween 20. For *A. appalachense*, chemical fixation did not yield good expansion, so Cryo-ExM was adapted from previously described protocols[62,63]. Briefly, cells were high-pressure frozen, followed by overnight freeze substitution in acetone with 0.25% formaldehyde and 0.05% glutaraldehyde on a metal block chilled in liquid N₂ placed in dry ice. Next day, this was followed by stepwise rehydration with 100%, 90%, 75% and 50% ethanol and finally the samples were resuspended in PBS. Samples were then crosslinked in solution on a shaker overnight to maximize crosslinking, following which gels were prepared, labelled and imaged as mentioned above.

### Light microscopy

For immunolabelled cells, we used poly-ʟ-lysine-coated Ibidi chamber slides (eight-well, Ibidi 80826). The wells were filled with 1× PBS and

0.4 μM Hoechst 33342 (ThermoFischer 62249) was added. Immunostained cells were added to wells and allowed to settle for an hour before imaging. Imaging was done on the Zeiss LSM 880 using the Airyscan Fast mode using the Zen software with the Plan-Apochromat 63×/1.4 Oil DIC M27 objective. For staining efficiency, sample overviews were imaged in LSM mode using the tilescan function with the Plan-Apochromat 63×/1.4 Oil DIC M27 objective. For immunolabelled U-ExM gels, we also used poly-L-lysine-coated Ibidi chamber slides (two-well, Ibidi 80286; four-well, Ibidi). Gels were cut to an appropriate size to fit the Ibidi chambers and added onto the wells. The gels were overlaid with water to prevent drying or shrinkage during imaging. The gels were imaged using the Zeiss LSM 880 with the Airy fast mode using a Plan-Apochromat 63×/1.4 Oil DIC M27 or an upright Leica SP8 confocal microscope with an HC PL APO 40×/1.25 glycerol objective or Nikon-CSU-W1 Sora with a SR P-Apochromat IR AC 60× WI/ 1.27 objective.

### Analysis of *S. arctica* MTOC duplication

In hydroxyurea inhibition assays (Extended Data Fig. 3), synchronized *S. arctica* cultures were grown at 17 °C as mentioned above in marine broth for 16 h. At this timepoint, one sample was fixed (16 h control) and 50 mM hydroxyurea was added to the second sample and grown for a further 8 h (hydroxyurea treated) and fixed at 24 h after inoculation. An extra untreated control sample was fixed at 24 h. Cells were immunostained for tubulin and DNA as above and cells were imaged. Interphase cells were classified as one or two MTOCs. All mitotic cells were counted as having one MTOC.

### Effect of inhibitor treatments on *S. arctica* mitosis

For microtubule perturbation in mitosis (Extended Data Fig. 6), acute treatment of low-concentration carbendazim (378674; Sigma) was used. *S. arctica* cells were grown as described above up to 28 h after synchronization, followed by the addition of 0.5 μg ml$^{-1}$ of MBC for 15 min and collected and fixed as above. For analysing the impact of microtubule depolymerization on NPC arrays in *S. arctica* (Extended Data Fig. 7), 24 h after synchronization, cells were treated with 25 μg ml$^{-1}$ of MBC for 4 h and collected and fixed as above. For impact of lipid depletion on *S. arctica* nuclei and mitosis (Extended Data Fig. 6), cells were treated with 25 μg ml$^{-1}$ cerulenin (CAS 17397-89-6; Santa Cruz Biotechnology) 6 h after synchronization. Cells were incubated for another 24 h before they were fixed.

### Analysis of mitosis in multinucleate *C. perkinsii* cells

In the cytokinesis blocking experiments (Extended Data Fig. 10), *C. perkinsii* synchronization was achieved through filtration. Subsequently, the cells were cultured at 23 °C for 45 h, followed by the addition of either dimethylsulfoxide or blebbistatin (reference 72402; StemCell Technologies) at a concentration of 1 mM. The cells were further incubated in the presence of the inhibitor for 30 h before fixation. Subsequent to fixation, U-ExM was used to visualize nuclei and microtubule spindles, facilitating the enumeration of cells exhibiting multipolar spindles. Cells were pan labelled with NHS-ester to demarcate cell boundaries.

### Image analysis

For immunostaining efficiency measurements (Extended Data Fig. 5a), immunostained (immunofluorescence) and expanded (U-ExM) samples were stained for microtubules (E7 antibody, DSHB). Cells were imaged using confocal microscopy in tilescan mode. Hoechst 33342 or NHS-ester was used as a reference to determine the percentage of immunostained cells.

For NPC density measurements (Fig. 2f), NPC density was determined using nuclei from U-ExM gels labelled with MAb414 (Biolegend 902901). In *S. arctica* NPC densities are different around the MTOC as compared to the rest of the nucleus. Thus, two regions of interest (ROIs) were selected per nucleus, one in the radial arrays in the vicinity of the MTOC and a second one away from it (marked in the graph as NE). Each ROI was a 5 μm cube. The nuclei were classified as interphase or mitotic, on the basis of nuclear shape and presence or absence of intranuclear microtubules. The images were thresholded and binarized. The three-dimensional (3D) object counter plugin was used to obtain NPC counts. The counts were divided by cube volume to obtain NPC density. The measurements were corrected for the gel expansion factor to obtain actual NPC density per μm$^3$.

For SPB dimensions (Extended Data Fig. 3b), analysis of SPB dimension was done using pan-labelled U-ExM gels. The images were cropped to a 5 μm region around the SPB, thresholded and binarized. The 'Analyse particle' function was used to obtain SPB shape measurements, including width and height (Extended Data Fig. 3b). The measurements were corrected for gel expansion factor to obtain actual width and height.

For *S. arctica* MTOC–MTOC distance (Fig. 2b), the distance was determined using tubulin-labelled U-ExM gels. The images were thresholded using morphological filtering and binarized. The structure was then skeletonized and the Analyse Skeleton plugin (https://github.com/fiji/AnalyzeSkeleton) was used to determine spindle length. In cases for which this was not possible, MTOC positions were marked manually and Euclidean distance was calculated between the two points.

*C. perkinsii* centriole diameter was measured in Fiji from TEM tomography images. Both longitudinal and transversely placed centrioles were used for the analysis. Centrioles were placed at varying angles to the sectioning and imaging plane.

We performed image analysis using Fiji software[64,65]. All figures were assembled with Illustrator 2022 CC 2020 (Adobe). Graphs were generated using GraphPad Prism 9. The 3D reconstructions of Supplementary Videos 3, 4, 7 and 8 were done in Imaris v.992.

### Live cell imaging

Light-sheet microscopy in Supplementary Video 2 was performed using the LS1 Live 246 light-sheet microscope system (Viventis) as previously described using a 25 × 247 1.1 NA objective (CFI75 Apo 25XW; Nikon) and an sCMOS camera (Zyla 4.1 andor)[31]. Light-sheet imaging was conducted in a room specifically cooled at 17 °C using an air-conditioning unit.

### Electron microscopy

A combination of three different sample preparation techniques was tried for the different ichthyosporean species and the one that yielded better results for each sample was chosen for further imaging and presented here.

**Sample preparation 1.** This applies to Figs. 1c, 2g (bottom) and 3f. For TEM tomography of *S. arctica* and *C. perkinsii* cells, samples were concentrated by sedimentation and high-pressure frozen with the HPM010 (Abra Fluid) using 200-μm-deep, 3-mm-wide aluminium planchettes (Wohlwend GmbH). Freeze substitution (FS) was done using the AFS2 machine (Leica microsystems) in a cocktail containing 1% OsO$_4$, 0.1% uranyl acetate and 5% water in acetone. The samples were incubated as follows: 73 h at −90 °C, temperature increased to −30 °C at a rate of 5 °C h$^{-1}$, 5 h at −30 °C, temperature increased to 0 °C at a rate of 5 °C h$^{-1}$, 4 × 0.5 h rinses in water-free acetone at 0 °C. This was followed by Epon 812 (Serva) infiltration without BDMA (25%−3 h at 0 °C, 50% overnight at 0 °C; 50%−4 h at room temperature; 75%−4 h, 75% overnight, 100% 4 h (×2) and 100% overnight. This was followed by exchange with 100% Epon 812 with BDMA (4 h × 2, followed by overnight). After this the samples were polymerized in the oven at 66 °C for over 2 days. The samples were then cut using an ultramicrotome (Leica UC7) in 70 nm sections screened by two-dimensional (2D) TEM (Jeol 1400 Flash) to assess sample preparation. For TEM tomography (300 nm sections), sections were poststained with 2% uranyl acetate in 70% methanol (5 min, room temperature) and in Reynolds lead citrate (2 min, room temperature).

**Sample preparation 2.** This applies to Figs. 1d and 2g (top) and Supplementary Videos 1 and 5. For serial tomography and FIB-SEM of *S. arctica* cells, samples were concentrated and high-pressure frozen as mentioned above. Freeze substitution was done in the AFS2 machine (Leica microsystems) in a cocktail containing 1% $OsO_4$, 0.5% uranyl acetate and 5% water in acetone. The samples were incubated as follows: 79 h at −90 °C, temperature increased to −60 °C at a rate of 2 °C h$^{-1}$; 10 h at −60 °C, temperature increased to −30 °C at a rate of 2 °C h$^{-1}$; 10 h at −30 °C, temperature increased to 0 °C at a rate of 5 °C h$^{-1}$; 1 h at 0 °C. After this, the samples were rinsed in acetone and further incubated in 0.1% thiocarbohydrazide, 10% water in acetone for 30 min at room temperature, followed by 1% $OsO_4$ in acetone. After rinsing, the samples were infiltrated in Durcupan ATM (Sigma) and finally polymerized in a 60 °C oven for 72 h. The $OsO_4$ step and the infiltration were performed in a Biowave (Ted Pella). The samples were then sectioned using an ultramicrotome (Leica UC7) for serial section TEM tomography (300 nm sections). The sections were poststained with 2% uranyl acetate in 70% methanol (5 min, room temperature) and in Reynolds lead citrate (2 min, room temperature). Tomograms were acquired with a Tecnai F30 (ThermoFisher Scientific) using SerialEM[39] and reconstructed and joined with Imod Etomo. After this, a 70 nm section was collected and screened by 2D TEM (Jeol 1400 Flash) to target interphase and mitotic cells for FIB-SEM analysis. The samples were then mounted on a SEM stub using silver conductive epoxy resin (Ted Pella), gold sputter coated (Quorum Q150RS) and imaged by FIB-SEM. The acquisition was performed using a Crossbeam 540 or 550 (Zeiss) following the Atlas 3D workflow. SEM imaging was done with an acceleration voltage of 1.5 kV and a current of 700 pA using an ESB detector (1,100 V grid). Images were acquired at 5 × 5 nm$^2$ pixel size and 8 nm slices were removed at each imaging cycle. FIB milling was performed at 700 pA current. For segmentation and visualization we used 3DMod and Amira (v.2019.3 or 2020.1; ThermoFisher Scientific).

**Sample preparation 3.** This applies to Extended Data Figs. 9 and 10. For TEM tomography and FIB-SEM of *A. appalachense and C. limacisporum* cells, we adapted a protocol shown to be compatible with FIB-SEM in ref. 66. We concentrated and high-pressure froze the samples and FS them with 0.1% uranyl acetate in dry acetone. After 72 h of incubation at −90 °C, the temperature was increased to −45 °C at a speed of 2 °C h$^{-1}$ and then the samples were incubated in the uranyl acetate solution for an extra 10 h at −45 °C. The samples were then rinsed with pure acetone before infiltration with increasing concentrations of the resin Lowicryl HM20 (Polysciences), while increasing the temperature to −25 °C. The blocks were polymerized with ultraviolet for 48 h at −25 °C. Finally, the samples were mounted on stubs, prepared and imaged by FIB-SEM as described above.

## Twist analysis

To calculate spindle twist for *C. perkinsii*, Fiji Software (ImageJ)[64] was used to analyse microscopy images of horizontal spindles. Only images with both spindle poles in the same plane or in two consecutive planes apart in each direction of the *z*-stack were included in the analysis to prevent spindle tilt from affecting the spindle twist calculation. Horizontal spindles were transformed into a vertical orientation (end-on view) using a previously developed code written in R programming language in RStudio[67]. In the transformed stack, microtubule bundles and poles appear as blobs. The spindle poles are tracked manually using the Multipoint tool in ImageJ. Next, we used the previously developed optical flow method to calculate the twist[68] and presented the absolute values in the graph. The tracing of bundles and twist calculations were previously written in Python programming language using PyCharm IDE, with external libraries such as NumPy, scikit-image, Matplotlib, PIL, OpenCV and SciPy. The code and instructions are available at GitLab: https://gitlab.com/IBarisic/

detecting-microtubules-helicity-in-microscopic-3d-images. Twist values for RPE1 cells expressing CENP-A-GFP and centrin1-GFP were taken from ref. 68.

## Analysis of spindle length, width and interkinetochore distance

To measure spindle length, width and interkinetochore distance, the Line tool in Fiji Software (ImageJ)[64] was used. Length was measured by drawing a line from pole to pole of the spindle. In *C. perkinsii* the pole positions were determined visually as the outermost points of the spindle along the central spindle axis. In RPE1 cells expressing CENP-A-GFP and centrin1-GFP, length was measured by using the images from ref. 27 and a line was drawn from one centrosome to the other. Width in *C. perkinsii* was measured by drawing a line across the equatorial plane of the spindle, with the line ending at the outer edges of the spindle. Width in RPE1 cells expressing CENP-A-GFP and centrin1-GFP was measured by drawing a line across the equatorial plane of the spindle, with the line ending at the outer kinetochore pairs. Interkinetochore distance in *C. perkinsii* was measured by using the Rectangle tool in Fiji Software which was drawn between endings of *k*-fibres at the spindle midzone. Interkinetochore distance in RPE1 cells expressing CENP-A-GFP and centrin1-GFP was taken from ref. 27. It was not possible to measure the interkinetochore distance in *S. arctica* because of the tight microtubule bundles of the spindles and it was not possible to distinguish kinetochore microtubules.

## Analysis of the bridging fibre intensity

To measure the intensity of bridging fibres[69] we used the Square tool (ImageJ)[64]. In *C. perkinsii*, the position of the square when measuring bridging fibre intensity was on the fibre located between the endings of kinetochore fibres. As we did not have labelled kinetochores and could not determine where a single kinetochore fibre is, we put squares close to the end of kinetochore fibres to obtain values of kinetochore fibres together with bridging fibres ($I_{bk}$) (Extended Data Fig. 7a). Background was measured and subtracted as follows: $I_b = I_{b+bcg} − I_{bcg}$ for bridging fibres and $I_{bk} = I_{bk+bcg} − I_{bcg}$ for bridging fibres together with kinetochore fibres. The values for RPE1 cells expressing CENP-A-GFP and centrin1-GFP were taken from ref. 27.

## Statistics and reproducibility

Results are reported as mean ± s.d. Statistical parameters including the numbers of cells, nuclei or MTOCs analysed, *n* and statistical significance are reported in the figure legends. Statistical significance was calculated by Mann–Whitney *U*-test, Kruskal–Wallis or Student's *t*-tests. Asterisks in graphs indicate the statistical significance (*$P < 0.05$; **$P < 0.01$; ***$P < 0.001$; ****$P < 0.0001$). We performed statistical analysis in GraphPad Prism 9. All light microscopy and electron microscopy images are representative images obtained from three and two independent experiments, respectively.

## Reporting summary

Further information on research design is available in the Nature Portfolio Reporting Summary linked to this article.

## Data availability

Data associated with the study are available at https://doi.org/10.6084/m9.figshare.c.6639812 (ref. 70). All microscopy datasets generated for this study are available at https://www.ebi.ac.uk/biostudies/bioimages/studies/S-BIAD1306. A duplicate of the data is hosted at s3 storage bucket: https://s3.embl.de/shahnature2024 accessible with access key: rSmWsv6HGFcBOvMLFsKI; secret key: 9O64iOgCflVIy0FVVgNQvHb83SBIUBMUpVpHaCLP. Data can be accessed through minIO Client (https://min.io/docs/minio/linux/reference/minio-mc.html) or other s3 compatible services. Source data are provided with this paper.

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

**Acknowledgements** We thank S. Martin, P. Gönczy, J. Ellenberg, B. Baum, R. Heald, A. Chaigne, P. Pereira, S. Culley, I. Raote, T. Quail, H. Vu, F. Vincent, A. Erzberger, J. Elliot, N. Petridou, N. Banterle, M. Dorrity and all members of the laboratories of G.D., O.D. and Y.S. for comments on the manuscript and general feedback. We thank H. Suga for *C. limacisporum* and *C. perkinsii* cultures, K. Ramachandran for help with comparative genomics tools and M. Araújo for establishing membrane staining protocols in expanded ichthyosporean cells. We thank the Advanced Light Microscopy Facility at the European Molecular Biology Laboratory (EMBL), Carl Zeiss, the Electron Microscopy Core Facility at EMBL and the biological electron microscopy facility at EPFL for their support. O.D. and M.O. are funded by an Ambizione fellowship from the Swiss National Science Foundation (PZOOP3_185859). G.D., H.S., C.B., P.R. and Y.S. are funded by the European Molecular Biology Laboratory. G.D. is funded by the European Union (ERC, KaryodynEVO, 101078291). H.S. is supported by the EMBL Interdisciplinary Postdoctoral Fellowship (EIPOD4) programme under Marie Sklodowska-Curie Actions Cofund (grant agreement no. 847543). E.T. is supported by an NWO-Veni Fellowship (VI.Veni.202.223). M.T. has been supported by the 'Young Researchers' Career Development Project—training of doctoral students' of the Croatian Science Foundation. I.M.T. acknowledges the support of European Research Council (ERC Synergy grant, GA 855158), Croatian Science Foundation (HRZZ project IP2019-04-5967) and projects cofinanced by the Croatian Government and European Union through the European Regional Development Fund—the Competitiveness and Cohesion Operational Programme: QuantiXLie Center of Excellence (grant no. KK.01.1.1.01.0004) and IPSted (grant no. KK.01.1.1.04.0057).

**Author contributions** H.S. coconceived the project, designed and implemented all the experiments, acquired, analysed and interpreted the data (with the exception of the specific, individual contributions from other authors listed subsequently) and contributed to the drafting of the paper. M.O. contributed experimental data and analysis to Fig. 3a,c–e and Extended Data Figs. 4, 7a,d, 8, 9a,b and 10g–i. C.B. contributed experimental data and analyses to Fig. 2a,b. P.R. contributed advice on protocols, analysis and support to the experiments underlying Figs. 1c,d, 2g and 3f and Extended Data Figs. 3d, 7c, 9d,e and 10c,f. M.T. and I.M.T. analysed data for Fig. 3c–e and Extended Data Fig. 8. E.C.T. provided phylogenetic analysis for Fig. 1b and Extended Data Fig. 1 and advice on homology searches and gene tree reconstruction. Y.S., O.D. and G.D. cosupervised the project and provided advice on experimental design, implementation and analysis. O.D. carried out experiments for Figs. 3a,c–e and Extended Data Figs. 4, 7a,d, 8, 9a,b and 10g–i. O.D. and G.D. coconceived the project and led the drafting of the manuscript. All authors provided input during the manuscript drafting stage.

**Funding** Open access funding provided by European Molecular Biology Laboratory (EMBL).

**Competing interests** The authors declare no competing interests.

**Additional information**

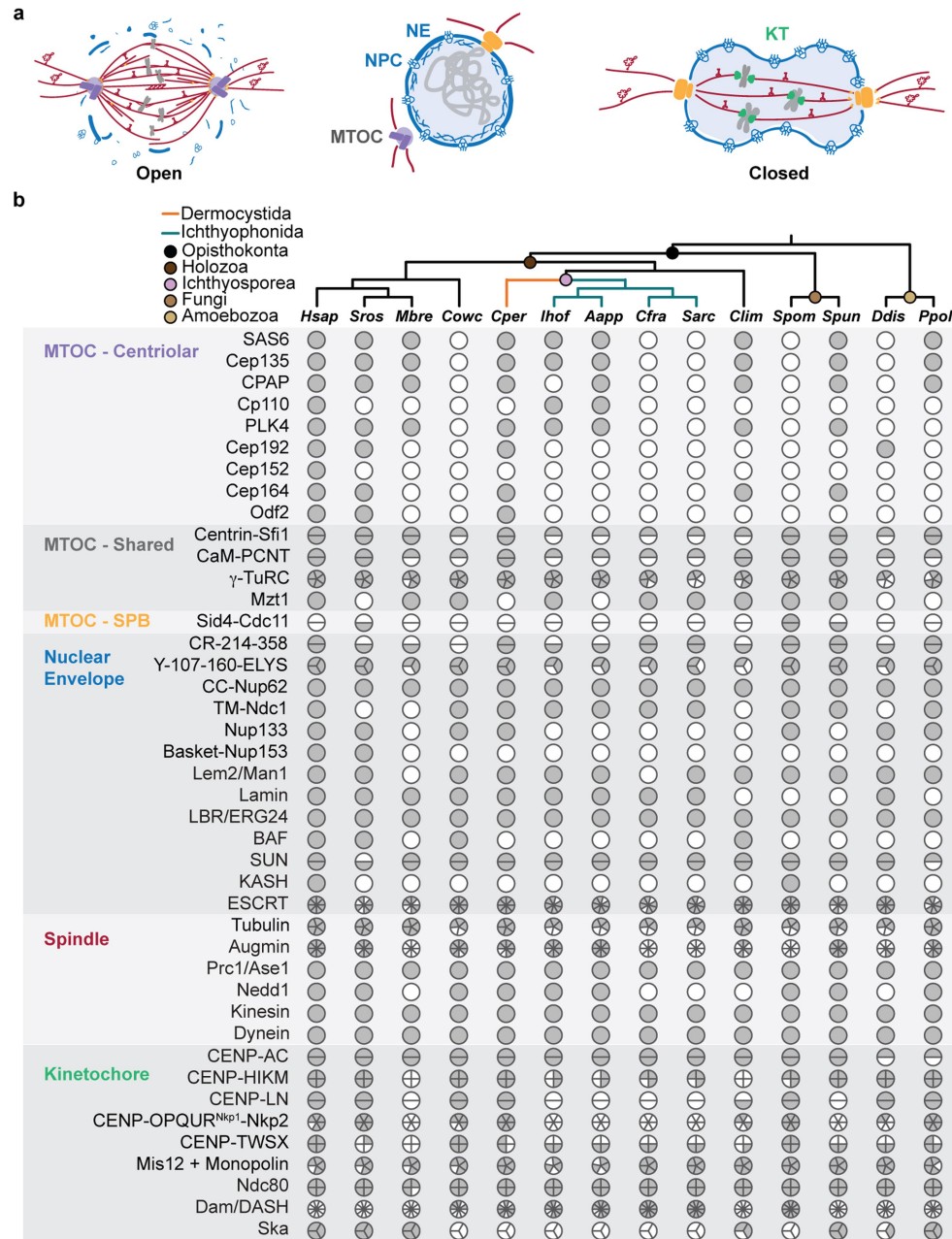

**Extended Data Fig. 1 | Divergence in mitosis-associated protein profile in close relatives of animals and fungi.** Phylogenetic profiles of proteins involved in mitosis. Filled and empty circles/pie charts indicate presence and absence of proteins, respectively (materials and methods). In addition to Ichthyophonida (*Sarc- S. arctica, Cfra- Creolimax fragrantissima, Aapp- Amoebidium appalachense* and *Ihof- Ichthyophonus hofleri*) and Dermocystida *Chromosphaera perkinsii* (*Cper*) and Corallochytrea *Corallochytrea limacisporum* (*Clim*), profiles of key species are represented including *Homo sapiens* (*Hsap*), *Schizosaccharomyces pombe* (*Spom*), the choanoflagellate *Salpingoeca rosetta* and *Monosiga brevicollis* (*Sros* and *Mbre*), the filasterean *Capsaspora owczarzaki* (*Cowc*), the early-branching chytrid fungus *Spizellomyces punctatus* (*Spun*) and two amoebozoan species *Dictyostelium discoideum* (*Ddis*) and *Physarum polycephalum* (*Ppol*).

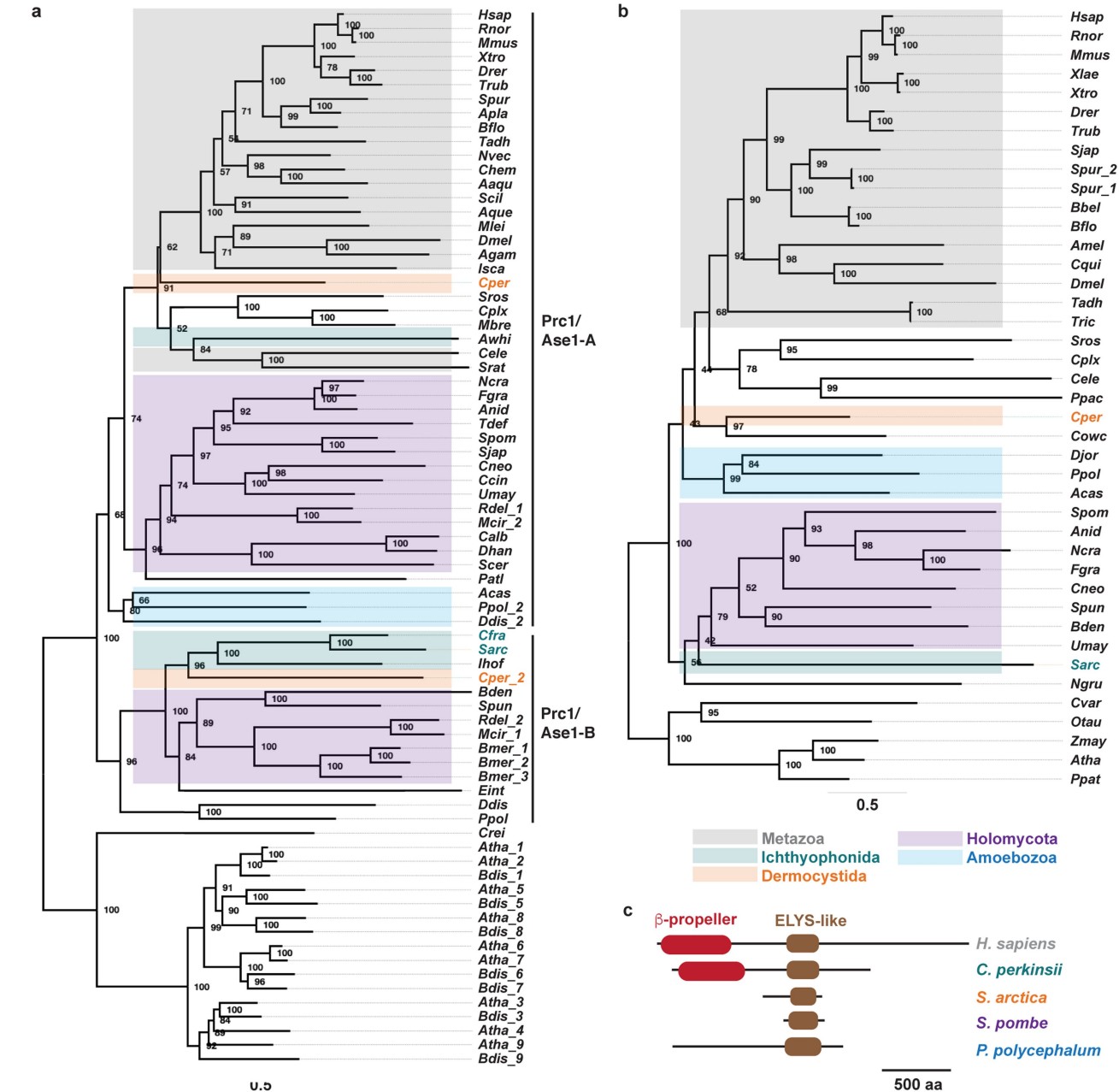

**Extended Data Fig. 2 | Divergence of selected mitotic proteins within Ichthyosporea. a**, In contrast to the A-type MT crosslinker protein PRC1/Ase1 present in most animal lineages, the ichthyophonid *S. arctica* retains the B-type protein, putting it on the opposite side of an early duplication. Both orthologs resulting from this duplication are found in amoebozoans, early-branching fungi and the dermocystid *C. perkinsii*. Trees were made using IQtree v2.1.2 with ultrafast bootstrap (1000) using the LG + IG + G4 model. **b**, Phylogenetic distribution of ELYS, involved in post-mitotic NPC reassembly. Trees were made using IQtree v2.1.2 with ultrafast bootstrap (1000) and LG + F + I + G4 model. The trees were visualised and annotated using FigTree (methods) Numbers on the nodes are bootstrap values. Branch length units are arbitrary. **c**, Dermocystid *C. perkinsii* has animal-like ELYS architecture with a central ELYS domain and an N-terminal β-propeller domain that has been lost in the fission yeast *S. pombe* as well as the ichthyophonid *S. arctica*.

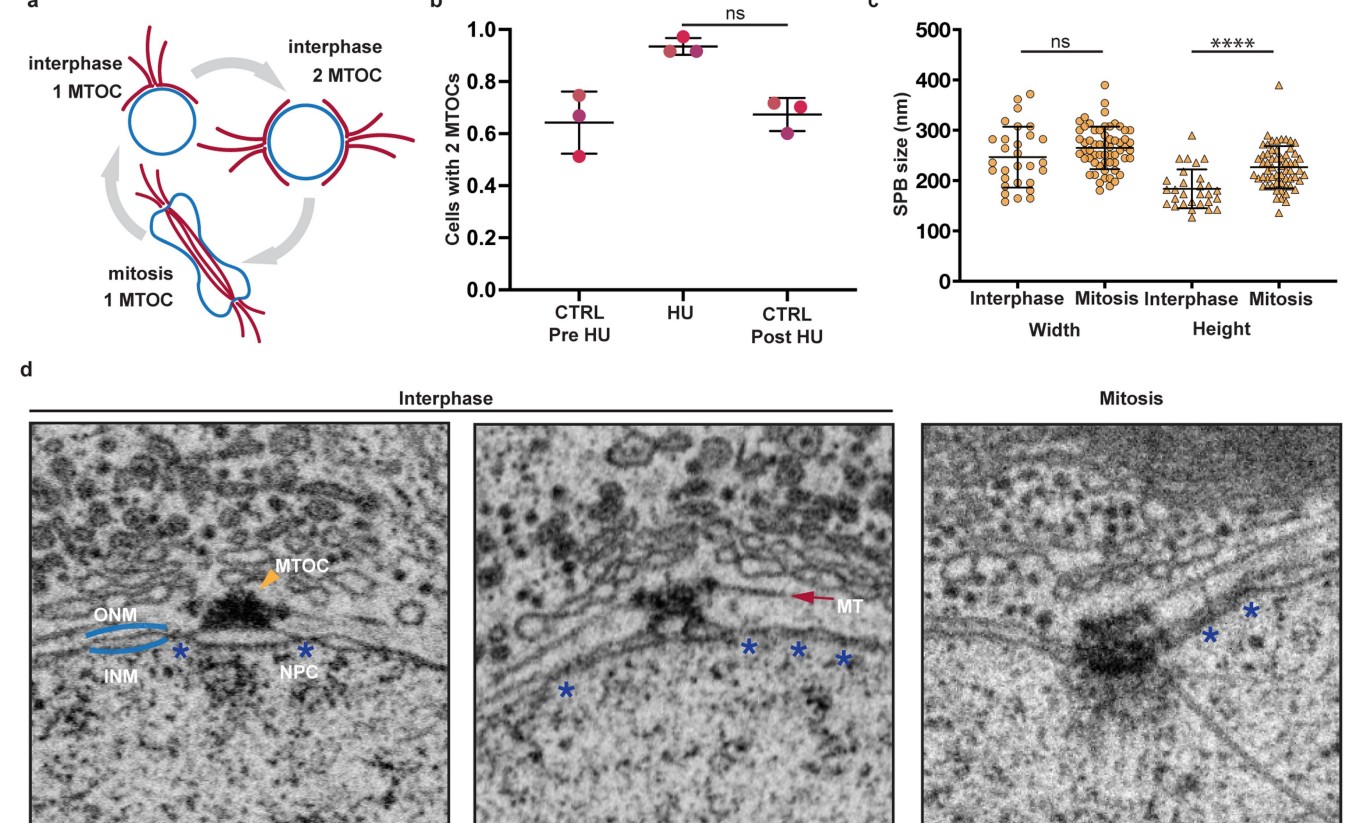

**Extended Data Fig. 3 | Duplication and NE embedding of the *S. arctica* MTOC. a**, Cartoon of *S. arctica* MTOC cycle, showing nuclei with 1 or 2 MTOCs. **b**, Dot plot showing proportion of cells with 2 MTOCs in control and hydroxyurea (HU) treated cells. The dots are shaded to indicate the three biological replicates ($n_{16h\_ctrl}$ = 121, 166 & 157, $n_{16h\_HU}$ = 108, 155 & 168, $n_{24h\_ctrl}$ = 111, 152 & 193). Statistical analysis was performed using a two-tailed Mann–Whitney *U*-test; *P* = 0.1000, not significant. Data are mean ± s.d. **c**, Dot plot showing dimensions of *S. arctica* MTOCs for nuclei in interphase and mitosis. Width and height of MTOCs were measured from NHS-ester labelled U-ExM gels ($n_{interphase}$ = 29, $n_{mitosis}$ = 60 MTOCs)

Statistical analysis was performed using a two-tailed Mann–Whitney *U*-test; Width-Interphase vs Mitosis *P* = 0.1021; Height-Interphase vs Mitosis *P* < 0.0001. Nuclei analysed over three biological replicates. Data are mean ± s.d. (ns-not significant; ****P* < 0.0001). **c**, Single slices through representative FIB-SEM volumes showing different configurations of the *S. arctica* MTOC-NE interface, from outer nuclear membrane (ONM) association in interphase to inner nuclear membrane (INM) embedding in mitosis. Blue asterisks indicate nuclear pore complexes (NPCs). Scale bars, 300 nm (**d**).

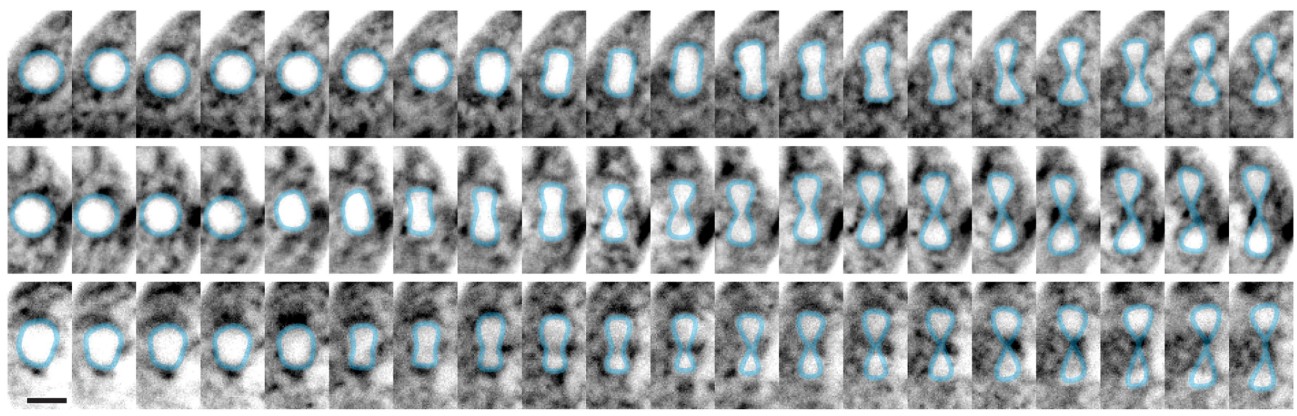

**Extended Data Fig. 4 | *S. arctica* goes through shape changes characteristic of closed mitosis.** *S. arctica* nuclei labelled with membrane marker FM4-64 and imaged by light sheet microscopy at 1 min intervals. Images are time-lapse montages of central slices through the nuclei. The NE is indicated in blue by manual annotation. Scale bar, 5 μm.

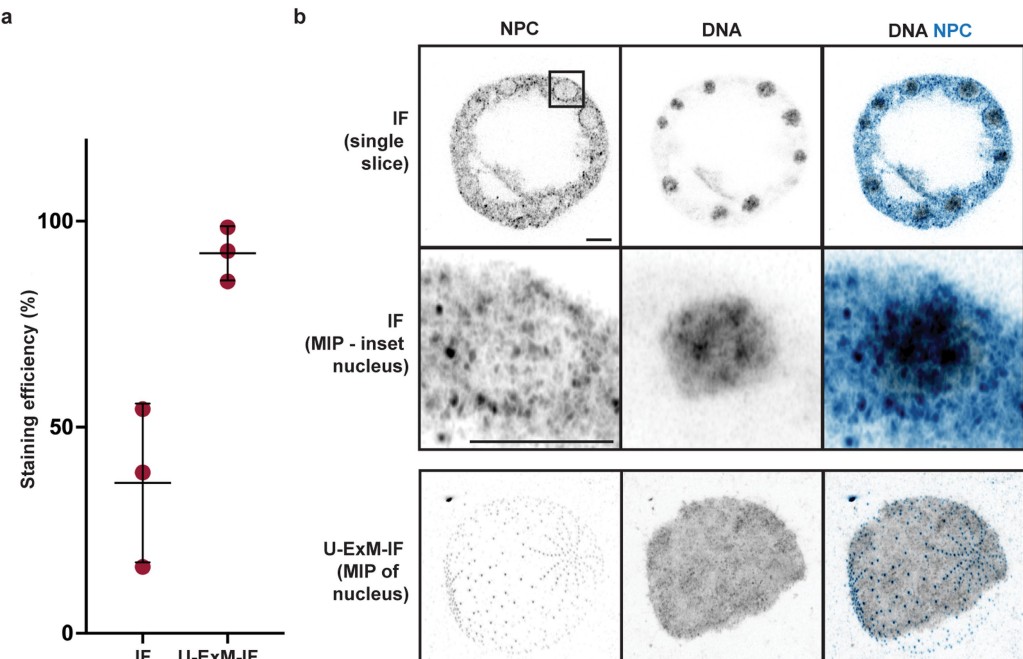

**Extended Data Fig. 5 | U-ExM improves immunostaining efficiency in Ichthyosporea. a**, Improvement in immunostaining efficiency when combined with U-ExM, measured as a percentage of cells labelled for tubulin (E7 antibody, DSHB) in three biological replicates; $n_{IF}$ = 199, 149, 164, $n_{UExM}$ = 134, 82, 69. Statistical analysis was performed using a two-tailed Mann–Whitney $U$-test; $P$ = 0.1000. Data are mean ± s.d. **b**, U-ExM reveals ultrastructural details of the *S. arctica* nucleus in comparison with classical IF. Maximum intensity projections (MIP) of representative *S. arctica* nuclei from three biological replicates stained either by IF (top) or IF post U-ExM (bottom) and labelled with MAb414 antibody (NPC, blue) and DNA (grey). Scale bars, 5 μm (**b**).

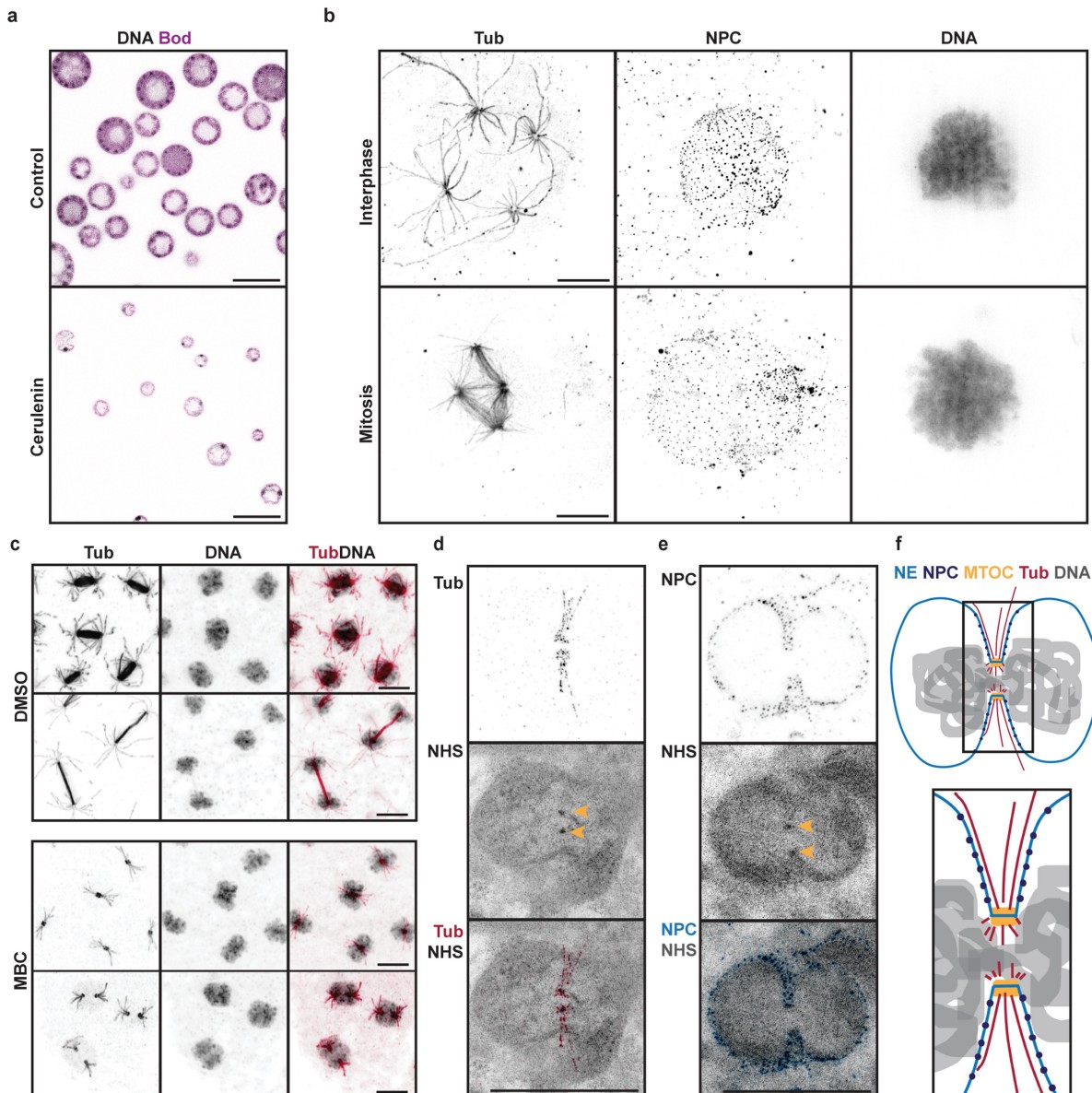

**Extended Data Fig. 6 | *S. arctica* nuclear remodelling in response to membrane and microtubule inhibition. a**, S. arctica cells arrest with cerulenin treatment. **b**, Maximum intensity projections of cerulenin-treated *S. arctica* interphase and mitotic nuclei immunolabelled for tubulin and NPCs. **c**, *S. arctica* spindles collapse with acute microtubule inhibition (0.5 μg/ml MBC for 15 min). Maximum intensity projections of representative nuclei of cells treated with DMSO and MBC and immunolabelled *S. arctica* spindles at different stages of mitosis. **d**-**e**, Maximum intensity projections of U-ExM images of *S. arctica* nuclei with tubulin (Tub - red) and pan-labelling (NHS) show loss of spindle MTs, while short astral MTs (**d**) and NPC radial arrays (blue) (**e**) still persist. **f**, Schematic representation of nuclear remodelling with increased polar indentation and spindle collapse in response to acute MT inhibitor treatment. Three independent experiments. Scale bars, 50 μm (**a**); 2 μm (**b**,**d**,**e**); 5 μm (**c**).

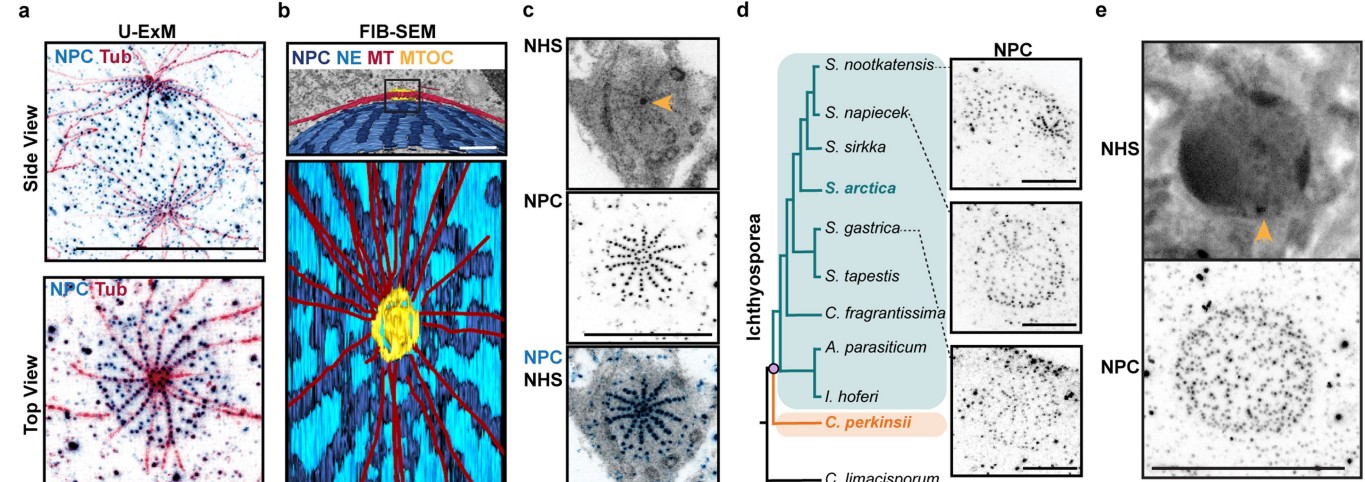

**Extended Data Fig. 7 | Radial distribution of NPCs surrounding the *S. arctica* MTOC. a**, Nuclear pore complexes (NPC- MAb414 antibody, blue) organised in radial arrays along MTs (tubulin, red). Images are representative maximum intensity projections of U-ExM stained *S. arctica* interphase nuclei from three biological replicates. **b**, 3D model of NPC arrays (dark blue) radiating from the MTOC (yellow) overlaid on an orthoslice of a representative *S. arctica* interphase nucleus imaged by FIB-SEM. **c**, NPCs are present at the centre of the MTOC in a subpopulation of nuclei. Representative maximum intensity projections of U-ExM images labelled with pan protein label NHS-ester (grey) and MAb414 antibody (NPC, blue). Yellow arrowhead indicates MTOC position. **d**, Cladogram of Ichthyosporea shows the position of *Sphaeroforma sp*. within the Ichthyophonida (teal) and neighbouring Dermocystids (orange). NPCs are organised in radial arrays across three distinct *Sphaeroforma* species. Images shown are representative maximum intensity projections of nuclei from *Sphaeroforma* sister species, *S. nootkatensis, S. napiecek* and *S. gastrica* are shown. **e**, NPCs are disorganised in *S. arctica* on prolonged microtubule inhibitor carbendazim (MBC) treatment (25 μg/ml MBC for 4 h). Images shown are maximum intensity projections of MBC-treated nuclei with pan protein labelling (NHS-ester) and MAb414 immunostaining. Yellow arrowhead indicates MTOC position. Scale bars, 5 μm (**a**,**c**,**e**); 500 nm (**b**); 2 μm (**d**).

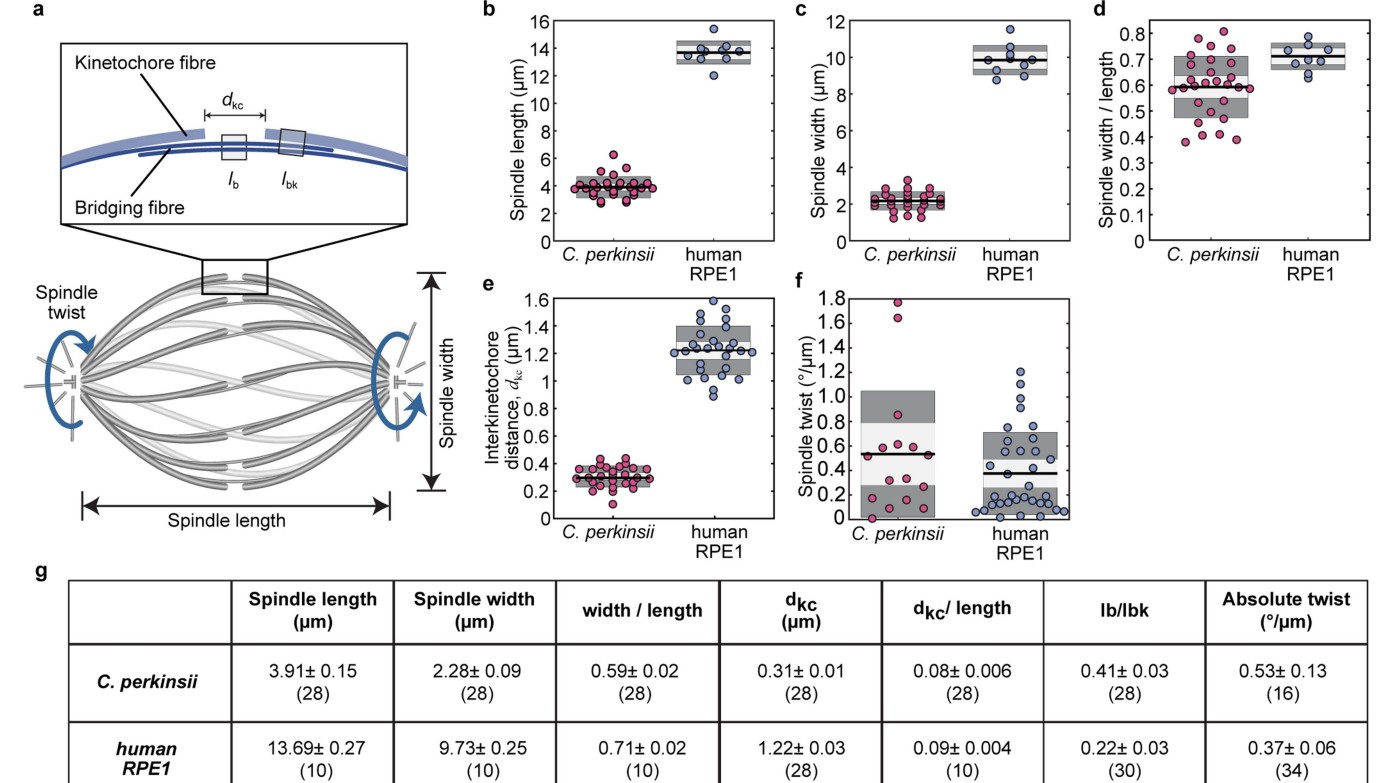

**Extended Data Fig. 8 | *C. perkinsii* shows features of human spindles.**
**a**, Schematic showing features of mitotic spindles including kinetochore and bridging fibres, interkinetochore distance ($d_{kc}$), twist, spindle length and width. **b-f**, Plot of spindle length; $P < 0.0001$, $n_{Cperk} = 28$, $n_{RPE1} = 10$ (**b**), spindle width; $P < 0.0001$, $n_{Cperk} = 28$, $n_{RPE1} = 10$ (**c**), spindle width/length ratio; $P = 0.00015$, $n_{Cperk} = 28$, $n_{RPE1} = 10$ (**d**), interkinetochore distance; $P < 0.0001$, $n_{Cperk} = 28$, $n_{RPE1} = 28$ (**e**) and absolute value of spindle twist; $P = 0.2742$, $n_{Cperk} = 16$,

$n_{RPE1} = 34$ (**f**) in *C. perkinsii* and human RPE1 cells. Data for RPE1 cells in (**d**) and (**e**) was taken from Štimac et al.[27]. Statistical analysis was performed using two-tailed Student's *t*-test. Three independent experiments. Data are mean (black line) ± s.d (dark grey). The light grey area marks the 95% CI on the mean. **g**, Table summarizing spindle parameters (**b-f**) in *C. perkinsii* and human RPE1 cells. n is indicated in brackets.

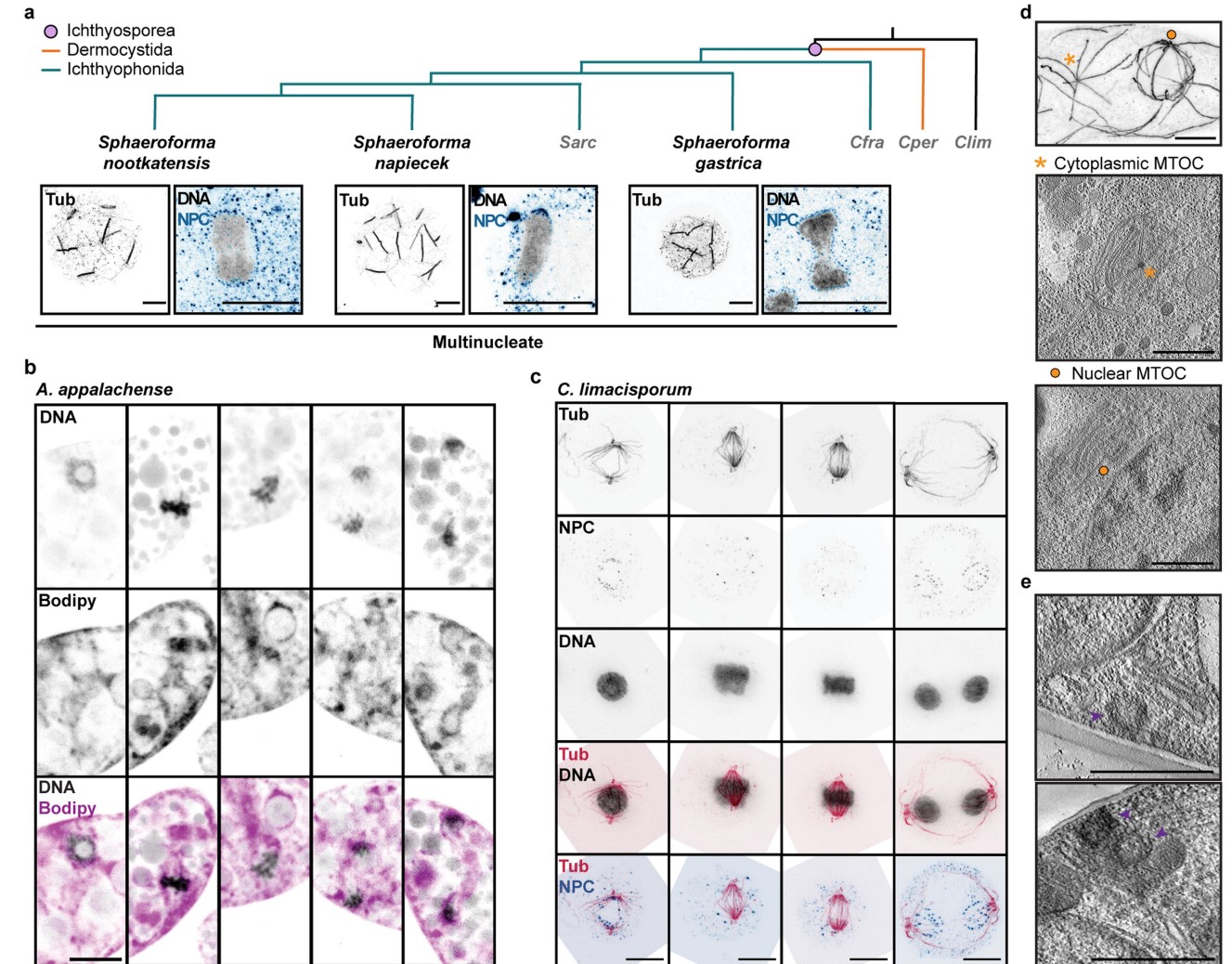

**Extended Data Fig. 9 | Mitotic strategies in diverse Ichthyosporean and corallochytrean species.** a, Cladogram shows phylogenetic relationship of *Sphaeroforma* species to other Ichthyophonida (teal) and Dermocystida (orange) within the Ichthyosporea. *Sphaeroforma* species undergo closed mitosis. Central panel shows representative maximum intensity projections of U-ExM images showing anaphase spindle architecture (left, Tubulin) and NPC organisation (right, NPC (blue) and DNA (grey)) in cells of representative *Sphaeroforma* species. **b**, Representative maximum intensity projections of mitosis in *A. appalachense* cells labelled for DNA (grey) and cellular membranes (magenta). **c**, Representative maximum intensity projections of *C. limaciporum* nuclei through mitosis, labelled microtubules, DNA and NPCs. **d**, *A. appalachense* has two distinct acentriolar MTOCs, cytoplasmic (yellow asterisk) and nuclear-associated (yellow circle). Single slice from a representative TEM tomogram of *A. appalachense* interphase nucleus **e**, *C. limacisporum* has a centriolar MTOC. Single slices from TEM tomography of *C. limacisporum* cells showing centrioles (purple arrows). Scale bars, 5 μm (**a**), 2 μm (**b** and **c**), 500 nm (**d** & **e**).

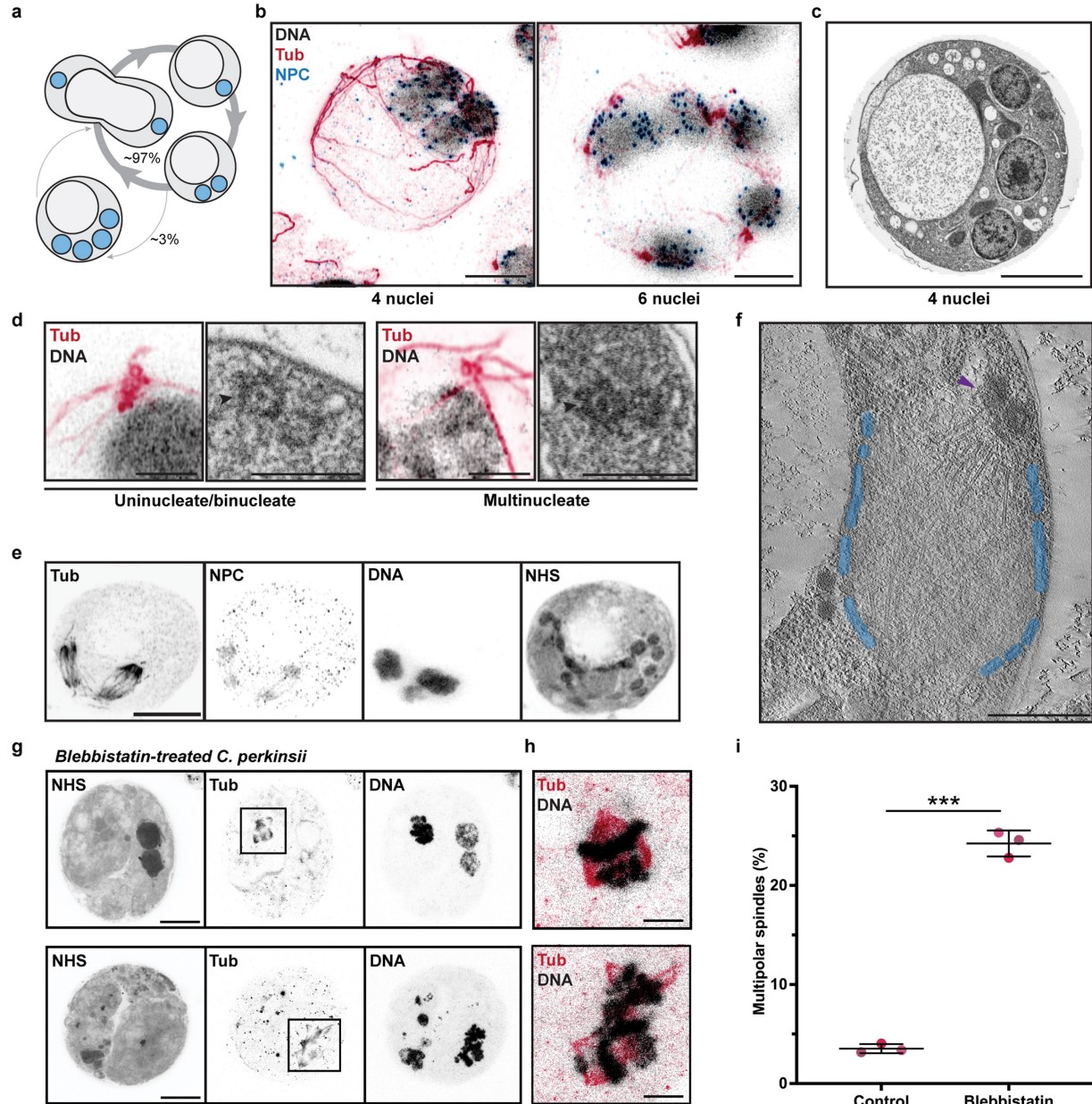

**Extended Data Fig. 10 | Mitosis in multinucleate *C. limacisporum* and *C. perkinsii*. a**, Life cycle of *C. limacisporum* showing the predominant uninucleate and binucleate states. **b**, Organization of multinucleate cells labelled for tubulin and DNA and pan-labelled with NHS ester in representative cells from three biological replicates. **c**, Single slice of a FIB-SEM volume showing ultrastructural organization of multinucleate cell. **d**, U-ExM (left) and FIB-SEM (right) images of centriolar MTOCs (black arrowheads) of uninucleate and multinucleate cells. U-ExM images are maximum intensity projections of cells labelled for tubulin. **e**, Synchronous mitosis in a multinucleate *C. limacisporum* cell labelled for tubulin and DNA and pan-labelled with NHS ester. **f**, TEM tomography image of *C. limacisporum* mitotic cell, nuclear number unknown, NE (blue) and centrioles (purple arrowhead) are marked.

**g**, Representative maximum intensity projections of multipolar spindles in *C. perkinsii* cells from three biological replicates treated with actin inhibitor blebbistatin and labelled for tubulin and DNA. Pan-labelling with NHS-ester indicates cell boundaries. **h**, Zoomed in images of insets marked in g. Merged maximum intensity projections of spindles labelled for DNA and tubulin. Images shown are maximum intensity projections of central slices through the nucleus. **i**, Dot plot showing proportion of cells with multipolar spindles in control and blebbistatin treated cells. The dots are shaded to indicate three independent replicates ($n_{control}$ = 99, 95 & 89, $n_{bleb}$ = 79, 61 & 75). Statistical analysis was performed using a two-tailed Student's *t*-test; *P* = 0.0004, df = 2.486. Data are mean ± s.d. (***P < 0.001). Scale bars, 2 μm (**b**; **c**; **d** (left); **e** & **h**), 500 nm (**d** (right) & **f**), 5 μm (**g**).

# Reporting Summary

Please do not complete any field with "not applicable" or n/a.  Refer to the help text for what text to use if an item is not relevant to your study.
For final submission: please carefully check your responses for accuracy; you will not be able to make changes later.

## Statistics

For all statistical analyses, confirm that the following items are present in the figure legend, table legend, main text, or Methods section.

| n/a | Confirmed | |
|---|---|---|
| ☐ | ☒ | The exact sample size (*n*) for each experimental group/condition, given as a discrete number and unit of measurement |
| ☐ | ☒ | A statement on whether measurements were taken from distinct samples or whether the same sample was measured repeatedly |
| ☐ | ☒ | The statistical test(s) used AND whether they are one- or two-sided<br>*Only common tests should be described solely by name; describe more complex techniques in the Methods section.* |
| ☒ | ☐ | A description of all covariates tested |
| ☒ | ☐ | A description of any assumptions or corrections, such as tests of normality and adjustment for multiple comparisons |
| ☐ | ☒ | A full description of the statistical parameters including central tendency (e.g. means) or other basic estimates (e.g. regression coefficient) AND variation (e.g. standard deviation) or associated estimates of uncertainty (e.g. confidence intervals) |
| ☐ | ☒ | For null hypothesis testing, the test statistic (e.g. *F*, *t*, *r*) with confidence intervals, effect sizes, degrees of freedom and *P* value noted<br>*Give P values as exact values whenever suitable.* |
| ☒ | ☐ | For Bayesian analysis, information on the choice of priors and Markov chain Monte Carlo settings |
| ☒ | ☐ | For hierarchical and complex designs, identification of the appropriate level for tests and full reporting of outcomes |
| ☒ | ☐ | Estimates of effect sizes (e.g. Cohen's *d*, Pearson's *r*), indicating how they were calculated |

*Our web collection on statistics for biologists contains articles on many of the points above.*

## Software and code

Policy information about availability of computer code

| Data collection | All data was collected on commercial microscopes running the following acquisition software: Zen (Zeiss), SerialEM and Atlas 3D. |
|---|---|
| Data analysis | Image analysis was performed in Fiji running ImageJ and Imaris (v992).  Additional analyses were carried out using HMMER suite (3.3.2), MAFFT (7.490), TrimAL (1.2), IQTree (2.1.2), iTOL (v5)GraphPad 9, RStudio and PyCharmIDE. Segmentation and visualisation of EM images was done with 3DMod and Amira (version 2019.3 or 2020.1). |

For manuscripts utilizing custom algorithms or software that are central to the research but not yet described in published literature, software must be made available to editors and reviewers. We strongly encourage code deposition in a community repository (e.g. GitHub). See the Nature Portfolio guidelines for submitting code & software for further information.

## Data

Policy information about availability of data

All manuscripts must include a data availability statement. This statement should provide the following information, where applicable:
- Accession codes, unique identifiers, or web links for publicly available datasets
- A description of any restrictions on data availability
- For clinical datasets or third party data, please ensure that the statement adheres to our policy

Data availability: Data associated with the study is available at https://doi.org/10.6084/m9.figshare.c.6639812 and additional bulk microscopy datasets are available in s3 storage bucket: shahnature2024 accessible with

## Research involving human participants, their data, or biological material

Policy information about studies with [human participants or human data](). See also policy information about [sex, gender (identity/presentation), and sexual orientation]() and [race, ethnicity and racism]().

| | |
|---|---|
| Reporting on sex and gender | NA |
| Reporting on race, ethnicity, or other socially relevant groupings | NA |
| Population characteristics | NA |
| Recruitment | NA |
| Ethics oversight | NA |

Note that full information on the approval of the study protocol must also be provided in the manuscript.

## Field-specific reporting

Please select the one below that is the best fit for your research. If you are not sure, read the appropriate sections before making your selection.

☒ Life sciences ☐ Behavioural & social sciences ☐ Ecological, evolutionary & environmental sciences

For a reference copy of the document with all sections, see [nature.com/documents/nr-reporting-summary-flat.pdf]()

## Life sciences study design

All studies must disclose on these points even when the disclosure is negative.

| | |
|---|---|
| Sample size | All data in this study represented either qualitative or quantitative analyses extracted from unmodified microscopy images. For fluorescence and/or electron microscopy images minimum of 5-10 cells per condition/timepoint per species were obtained. Following initial analysis, additional analysis was collected for any condition with notably lower counts. Sample sizes were determined to enable rigorous testing of hypotheses raised by preliminary experimental data. Data acquisition was continued until further acquisition was causing no measurable difference in results. |
| Data exclusions | No data were excluded from the analyses. |
| Replication | All experiments were repeated with two or three independent biological replicates tested on different days with different source cultures, as specified in legends and methods section. A proportion of the experiments were repeated in two different labs. A proportion of the imaging assays were replicated by a postdoc in the lab, albeit at a different stage of development. All attempts at replication were successful. |
| Randomization | This is not applicable for our study. All microscopy images were obtained randomly from synchronized cell populations at different days and in some cases across two distinct laboratories. |
| Blinding | Data were analysed where possible by automated or semi-automated methods. In cases where manual filtering or correction was incorporated, same criteria were applied to all strains or treatments under investigation. |

## Reporting for specific materials, systems and methods

We require information from authors about some types of materials, experimental systems and methods used in many studies. Here, indicate whether each material, system or method listed is relevant to your study. If you are not sure if a list item applies to your research, read the appropriate section before selecting a response.

## Materials & experimental systems

| n/a | Involved in the study |
|-----|----------------------|
| ☐ | ☒ Antibodies |
| ☒ | ☐ Eukaryotic cell lines |
| ☒ | ☐ Palaeontology and archaeology |
| ☐ | ☒ Animals and other organisms |
| ☒ | ☐ Clinical data |
| ☒ | ☐ Dual use research of concern |
| ☒ | ☐ Plants |

## Methods

| n/a | Involved in the study |
|-----|----------------------|
| ☒ | ☐ ChIP-seq |
| ☒ | ☐ Flow cytometry |
| ☒ | ☐ MRI-based neuroimaging |

## Antibodies

| | |
|---|---|
| Antibodies used | Anti-Tub E7 antibody (DSHB), Anti-Tub (NB600-936 Novus Biologicals), AA344 and AA345 (ABCD antibodies), Anti-NPC MAb414 (Biolegend 902901), Goat anti-Mouse Secondary Antibody, Alexa Fluor 488(Thermo A-11001) , Goat anti-guinea pig Secondary Antibody, Alexa Fluor 568 (Thermo A-11075), Goat anti-rabbit Secondary Antibody, Alexa Fluor 568 (Thermo A78955). |
| Validation | All primary antibodies used in the study are well established in the field and frequently used for immunofluorescence of the targets. All antibodies were supported by manufacturer's validation statements for purposes utilised in the study.<br><br>Anti- Tub E7 was previously validated in various single-celled eukaryotes :<br>Choanoflagellates : https://www.sciencedirect.com/science/article/pii/S0960982213011275?via%3Dihub , https://journals.plos.org/plosbiology/article?id=10.1371/journal.pbio.3000226<br>Leishmania chagasi: https://www.sciencedirect.com/science/article/pii/S0021925820890899?via%3Dihub<br><br>Anti-Tub (NB600-936 Novus Biologicals) was cross-referenced against Anti-TubE7 in our experiments.<br><br>Anti-NPC MAb414 (Biolegend 902901) was previously validated in various single-celled eukaryotes :<br>Yeast : https://journals.biologists.com/jcs/article/135/24/jcs260240/286062/Ultrastructure-expansion-microscopy-reveals-the<br>Theleiria parasite: https://journals.asm.org/doi/10.1128/msphere.00709-19<br>Plasmodium berghei : https://www.ncbi.nlm.nih.gov/pmc/articles/PMC9601220/ |

## Animals and other research organisms

Policy information about studies involving animals; ARRIVE guidelines recommended for reporting animal research, and Sex and Gender in Research

| | |
|---|---|
| Laboratory animals | The study did not involve laboratory animals- only lab strains of the unicellular models Ichthyosporeans (Sphaeroforma sp. Crolimax fragrantissima, Amoebidium sp. ) and Corralochytrium sp. |
| Wild animals | This study did not involve wild animals |
| Reporting on sex | For non-animal model organisms, reporting on sex is not relevant. |
| Field-collected samples | This study did not involve field-collected samples |
| Ethics oversight | Ethics approval is not required for research using unicellular holozoa. |

Note that full information on the approval of the study protocol must also be provided in the manuscript.

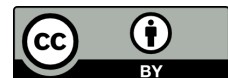

