## [Peer Review File · Nature]

Manuscript Title: Life cycle-coupled evolution of mitosis in close relatives of animals

Reviewer Comments & Author Rebuttals

Reviewer Reports on the Initial Version:

Referees' comments:

Referee #1 (Remarks to the Author):

The authors pose a central question in the field of molecular cell biology: What factors have influenced the evolution towards either closed or open mitosis? While both solutions are prevalent in eukaryotes, there is currently no clear phylogenetic pattern indicating a preference for one over the other.

This paper focuses on an understudied group of organisms, known as Ichtyosporea, which are deep-branching ophistokonts. Interestingly, the two lineages within Ichtyosporea, the Dermocystida and Ichthyophonida, exhibit distinct approaches to mitosis. Dermocystida are uninucleate and undergo open mitosis, while Ichthyophonida are multinucleate and undergo closed mitosis.

To compare the mitotic strategies of these groups, the authors employ advanced comparative phylogenomics along with high-end imaging techniques, primarily using *C. perkinsii* and *S. arctica* as representative organisms. This unique approach yields compelling data supporting the main conclusion of the manuscript: a multinucleated lifestyle necessitates closed mitosis, whereas a uninucleated lifestyle favors open mitosis.

The overall quality of the paper is excellent.

However, I have a few minor comments:

(1) The density of information, particularly in the figures, makes the manuscript challenging to read and navigate. For example, I doubt that a general audience would be able to fully grasp the significance of Figure 1. The MIPs in 1a are tiny, and one must rely solely on the legend to understand what is expected to be seen. Additionally, the filled circles in 1b appear more orange than yellow. The images in 1c and 1d, while visually appealing, do not contribute substantially to the findings. Furthermore, a link to the extended data is missing.

(2) I would like to suggest including the two members of Amoebozoa, *Dictyostelium* and *Physarum*, in Figure 1b, as has been done in Figure 4 and extended Figure 1.

(3) The paper primarily focuses on ophistokonts as this group contains the human and yeast models. However, this approach neglects the larger portion of the eukaryotic tree. It would be good to know if the authors postulate that their findings are universally relevant.

(4) In the regard, Physarum is a good example, as it can do both, open and closed mitosis. Although the work is cited (52)(citation 53 is incorrect, by the way), the basic finding by Solnice-Krezel et al. (1991) is the same as in the present manuscript. I find the discussion about the development of the Physarum solution a little trivialising.

(5) Figure 3c is not clearly explained in the legend, and its significance is not mentioned in the main text.

In summary, I commend the authors on their exceptional work.

Referee #2 (Remarks to the Author):

In this manuscript, Shah and colleagues address the evolution of mitotic mechanisms, and more specifically the choice between open and closed mitosis, focusing on a group of unicellular eukaryotes closely related to metazoans. Both strategies have a wide distribution and are sometimes found in the same species, and the question of the conditions that favor one or the other has long been posed. This manuscript presents the characterization of two species of Ichthyosporea, representing the two known lineages, Dermocystida and Ichthyophonida. By searching their genomes for orthologs of genes involved in different aspects of mitosis, the authors identify patterns of gene conservation or loss that suggest differences in the mitotic strategies of the two species.

Chromosphaera perkinsii possesses genes known to encode centriole components that *Spheroforma arctica* does not. This is confirmed by electron microscopy of both species, which reveals the presence of centrioles in the former and a structure reminiscent of the SPB of higher fungi in the latter. These analyses also show that, like higher fungi, *S. arctica* uses closed mitosis. In contrast, *C. perkinsii* displays an open mitosis similar to that observed in animal cells: the nuclear envelope breaks down, the spindle is organized by centrioles, and the chromosomes condense to form an equatorial plate. The properties of the spindle (twisting, scaling of inter-kinetochore distance) are comparable to those of human cells. The species most closely related to the Ichthyosporea, *Corallochytrium limacisporum*, also exhibits open mitosis. The authors conclude that open mitosis was probably present in a common ancestor of Ichthyosporea and Corallochytria, and thus pre-existed metazoans. They also conclude that the evolution of closed mitosis in Ichthyosporea and probably in other phyla as well is linked to the presence of a multinucleate stage in the life cycle.

This study is of great interest for understanding the evolutionary origins of Holozoa, to which metazoans belong. The characterization of mitosis in two species belonging to the two major lineages of Ichthyosporea, and the demonstration of their resemblance to the mitoses of animals on the one hand, and fungi on the other, is an important contribution to understanding this evolutionary history. The work is well executed and overall adequately presented. Furthermore, this work brings considerable weight to the hypothesis of a link between closed mitosis and coenocytic development. However, some points need to be improved before it can be considered for publication in Nature.

First, the authors state that their “analyses produce several experimentally verifiable predictions, including the presence of centrioles in Dermocystida and acentriolar MTOCs in Ichthyophonida by an overall divergence in spindle architecture, NE and NPC dynamics, and kinetochore organisation between the two lineages”. If there are indeed differences in the conservation of some of the genes involved in these processes, it is not obvious how they are predictive, to the exception of the loss of centriole components whose signature is well characterized. For example, lamins are lost in Ichthyophonida, but also in *C. limacisporum*, which nevertheless uses open mitosis. Similarly, it is mentioned that only a truncated form of ELYS is present in *S. arctica*, but no ortholog is identified in *C. limacisporum*. Some of the CenPHIKMs are lost in Ichthyophonida (and not always the same ones), but they are all lost in *C. limacisporum*. Among the Ichthyophonida, *Ichthyophonus hofleri* has virtually all augmin subunits.

Apart from centrioles, the clearest variation is the loss of CenO-P-Q-R/Nkp1-U-Nkp2 - although Nkp1 and Nkp2 are also lost in *C. limacisporum* - and CenpN, but it's not clear how this can be linked to the

transition from open to closed mitosis. Other elements, such as the low chromosome condensation observed during mitosis in Ichthyophonida, could possibly also explain some of these variations. The fact that most of the variations in the genes highlighted by the authors do not point clearly or very convincingly to a change in mitotic strategy is probably due in part to the choice made in presenting the results. Another option would be to start from the observation that the two lineages use different mitotic strategies to identify genes that are lost or modified. Alternatively, the absence of centriolar genes, which is associated in other eukaryotes with the development of divergent MTOCs, would in itself justify a more detailed analysis of mitotic mechanisms. Once the difference between species in terms of mitotic strategies has been confirmed, further genome analysis can be carried out to determine how this affects the conservation of spindle components, the nuclear envelope, etc.

- *C. limacisporum* also appears to have a coenocytic stage (doi.org/10.1016/j.cub.2021.06.061), as do other Dermatocystida. Is mitosis closed, and what do MTOCs look like in these species? A life cycle comprising both a uninucleate and a multinucleate stage, with open or closed mitosis, respectively, could have pre-existed the divergence between Corallochytraea and Ichthyosporea. In this case, MTOCs resembling the *S. arctica* MTOC could also be present in Dermatocystida and in *C. limacisporum* during multinucleate stages. Open mitosis would have been lost in the Ichthyophonida - or only in some of them, since some of the gene losses observed in *S. arctica* are not seen in *Amoebidium parasiticum* and *Ichthyophonus hofleri*. The protocol for expansion microscopy combining tubulin and NPC labelling could help to better characterize these transitions. That said, there does not seem to be any consensus as to the greater phylogenetic proximity between *C. limacisporum* and the Ichthyosporea within the Holozoa (see doi.org/10.1016/j.cub.2017.06.006). What are the arguments in favor of the phylogeny presented here? It could indeed change the interpretation of the observations.

Other comments :

- When does the duplication of *S. arctica* MTOC occur? Have the authors observed any duplication intermediates?

- It is interesting to find some centriolar proteins (SAS6, CEP135, Plk4) but not all (CPAP, TUBE, TUBD) in *Amoebidium parasiticum* and *Ichthyophonus hofleri*. Could there be a simplified centriole in these species like for instance in *C. elegans*? And what about their life cycles: are they only dividing as coenocytes? Here again, expansion microscopy could help address these questions relatively easily.

- The image of *C. limacisporum* (Extended data Fig.8) is not very informative as such. An image of an interphase cell showing the NPCs should be added for comparison. Also, the image of *C. fragrantissima* is of poor quality, and the NPCs are not visible at all.

- Members of the Augmin complex are referred to as HAUS in Extended data Fig.1

-l101: "centromere associated network (CCAN) network".

Referee #3 (Remarks to the Author):

(A, B, C, E, H) In this manuscript, the authors have investigated the manner of mitosis in the clade Ichthyosporea, close relatives of animals. Dey and his co-workers have been studying cells of eukaryotes in order to know how (and why) they evolved two extreme strategies of cell division, namely, open and closed mitosis. They have recently found that fission yeast uses a surprisingly similar mechanism of nuclear remodeling to animals, even though it adopts closed mitosis strategy, in contrast to animals which mostly uses the open strategy. Their achievement has been outstanding, as they not only found a common molecular mechanism between the two different mitotic strategies, but also elucidated how eukaryotes keep the integrity of nuclear envelop during closed mitosis, in which the membrane would be highly expanded.

Now, they focus more on the divergence of the strategies, using Ichthyosporea which contains two closely-related clades that undergo open or closed mitosis. They have found that *S. arctica*, which proliferates as a coenocyte, adopts the closed strategy, using state of the art techniques including TEM tomography, the focused ion beam scanning electron microscopy, and antibody staining based on the ultrastructure expansion microscopy. In contrast, another ichthyosporean *C. perkinsii* undergoes open mitosis, in a surprisingly similar manner as humans. They have concluded that the momentum that drove Ichthyophonida towards closed mitosis would be the restriction from their life cycle, namely, the multi-nucleated development, which may require an additional setup to ensure the faithful segregation in a shared space. Their findings have strongly reinforced the previous notions obtained from *Drosophila* embryo, animal germ lines, hyphal growth of fungi, and some amoebozoans.

Their conclusion is supported by a line of solid evidence that were produce by amazingly well-developed microscopic techniques. Their work has provided an explanation how ichthyosporeans have adopted their strategies of mitosis along with the restriction from life cycle, filling the gap between previous studies on fungi and animals. However, I do not think this manuscript represents the greatest scientific significance, compared to their previous work on fission yeast. I recommend authors to publish it in a more specialized journal.

(D) [Statistics] Statistics are correctly used.

(F) [Improvement] None specifically. However, I feel that they need to experimentally prove their conclusion that the nuclear envelop of multi-nucleated organisms is indispensable for a faithful segregation of nuclear materials using, for example, genetically modified *S. arctica* (or other Ichthyophonida species) whose nuclear envelop are disturbed.

(G) [Reference] I don't find any missing reference. However, the authors need to clarify from where the *S. arctica* and *C. perkinsii* were obtained.

Author Rebuttals to Initial Comments:

RESPONSE TO REFEREES

We sincerely appreciate your time, effort and expertise. Your comments were insightful and constructive, and over the last months, we have made it our main priority to generate the data required to fully address your comments and concerns. The revised paper is, in our opinion, far stronger as a result of peer review. Thank you.

Throughout this response document, we use **blue text** to indicate our response, **black text** to indicate the referee's original comment, and **magenta text** to indicate a quote taken from the manuscript.

Referees' comments:

Referee #1 (Remarks to the Author):

The authors pose a central question in the field of molecular cell biology: What factors have influenced the evolution towards either closed or open mitosis? While both solutions are prevalent in eukaryotes, there is currently no clear phylogenetic pattern indicating a preference for one over the other.

This paper focuses on an understudied group of organisms, known as Ichthyosporea, which are deep-branching opisthokonts. Interestingly, the two lineages within Ichthyosporea, the Dermocystida and Ichthyophonida, exhibit distinct approaches to mitosis. Dermocystida are uninucleate and undergo open mitosis, while Ichthyophonida are multinucleate and undergo closed mitosis.

To compare the mitotic strategies of these groups, the authors employ advanced comparative phylogenomics along with high-end imaging techniques, primarily using *C. perkinsii* and *S. arctica* as representative organisms. This unique approach yields compelling data supporting the main conclusion of the manuscript: a multinucleated lifestyle necessitates closed mitosis, whereas a uninucleated lifestyle favors open mitosis.

The overall quality of the paper is excellent.

Thank you for the positive feedback; we are glad you find our work of interest!

However, I have a few minor comments:

(1) The density of information, particularly in the figures, makes the manuscript challenging to read and navigate. For example, I doubt that a general audience would be able to fully grasp the significance of Figure 1. The MIPs in 1a are tiny, and one must rely solely on the legend to understand what is expected to be seen. Additionally, the filled circles in 1b appear more orange than yellow. The images in 1c and 1d, while visually appealing, do not contribute substantially to the findings. Furthermore, a link to the extended data is missing.

To address this comment, we have completely re-organised Figure 1 in the following ways:

- A new panel, Figure 1A, now summarises the life cycles and relative phylogenetic placement of the key species in our study, with larger images for greater clarity. In this revision, we include *Amoebidium appalachense* (Sarre et al. 2024, <https://doi.org/10.1101/2024.01.08.574619>) for the first time, for genomic comparisons (Extended Data Figure 1) as well as experimental analysis (Figure 1, Figure 4, Extended Data Figure 8).
- Figure 1b has been streamlined. We have moved several protein families to the Extended Data; the phylogenetic profiles are now in grayscale to prevent confusion arising from too many colours; the mitotic mechanism schematic has been simplified.
- We have decided to retain Figures 1c and 1d as they provide a direct experimental validation of the phylogenetic profiling approach for a key component of the mitotic machinery: the analysis in 1b predicts a centriole for *C. perkinsii* and an acentriolar MTOC for *S. arctica*.
- A link to the extended data is now included in the legend for Figure 1 (text below) **b**, *Cladogram of opisthokonts, highlighting the position of Ichthyosporea between well-studied animal, fungal and amoebozoan model systems. Phylogenetic profiles of selected proteins involved in mitosis (complete profiles in Extended Data Fig. 1). Filled and empty circles or pie charts indicate the presence and absence of proteins respectively (methods).*

(2) I would like to suggest including the two members of Amoebozoa, Dictyostelium and Physarum, in Figure 1b, as has been done in Figure 4 and extended Figure 1.

This is a good suggestion. Both Amoebozoa, *D. discoideum* and *P. polycephalum*, are now included in Fig. 1b, keeping the list of species consistent between Figure 1b and Figure 4b. Extended Data Figure 1 includes additional holozoan species as before.

(3) The paper primarily focuses on opisthokonts as this group contains the human and yeast models. However, this approach neglects the larger portion of the eukaryotic tree. It would be good to know if the authors postulate that their findings are universally relevant.

We agree that this is a necessary clarification. In brief, we believe the central finding of the paper, that multinucleated life cycle stages drive species towards a more closed mitosis, to be universally relevant across the tree of eukaryotes. Additional observations drawn from this study and the literature, such as the fact that open mitosis within opisthokont species appears to correlate with the presence of centrioles, clearly do not extend to other groups of eukaryotes and likely reflect a specific physiological feature of the opisthokont ancestor.

We have updated the text to reflect this more explicitly. The relevant section is highlighted here:

Line 273: *The broad range of examples of closed mitosis with centrioles (Fig. 4)^{44–46}, and open mitosis without centrioles^{47,48} outside the Opisthokonta argues that the presence of a flagellum or basal body does not per se constrain the mode of mitosis. Instead, our results suggest that having a coenocytic life cycle stage, in which more than two nuclei must divide and faithfully segregate in a shared cytoplasm, requires a closed or semi-closed mitosis (Fig. 4 and Extended Data Fig. 9). This model could explain the semi-closed or closed mitosis observed in the *Drosophila* coenocytic embryo^{49,50}, the germline of various animal lineages^{51–53}, and hyphal fungi^{8,54}, and is likely broadly generalisable to other eukaryotes*

outside the Opisthokonta, as in apicomplexan parasites or the coenocyte of Physarum polycephalum^{43,45}. A corollary of our hypothesis is that once closed mitosis has evolved, it can persist even if the organism returns to a unicellular, uninucleate life cycle^{7,55} but in such cases is apparently no longer under strict selection to remain closed^{56,57}.

(4) In this regard, Physarum is a good example, as it can do both, open and closed mitosis. Although the work is cited (52)(citation 53 is incorrect, by the way), the basic finding by Solnice-Krezel et al. (1991) is the same as in the present manuscript. I find the discussion about the development of the Physarum solution a little trivialising.

We agree and have updated the discussion to better reflect the critical importance of studying Physarum to understand transitions in mitotic mechanisms across the open-closed spectrum.

Line 258: The centriole-dependent open mitosis of C. limacisporum (Fig. 4 and Extended Data Fig. 8 and 9) together with its limited capacity for coenocyte formation suggests that the holozoan ancestor was capable of open mitosis, feasibly as part of a complex life cycle alternating between uninucleated cells with centriole-dependent open mitosis and coenocytes with acentriolar closed mitosis. Such a life cycle would be closely analogous to that of Physarum polycephalum (Fig. 4), which transitions from a crawling uninucleate amoeba with centrioles and open mitosis to a giant acentriolar coenocyte carrying out a closed mitosis⁴³.

(5) Figure 3c is not clearly explained in the legend, and its significance is not mentioned in the main text.

Line 193: c, C. perkinsii spindle in metaphase coloured for depth as shown in the legend on the left (top); enlarged section of the spindle showing kinetochore and bridging fibres (bottom) analysed in d and e and Extended Data Fig. 7.

In summary, I commend the authors on their exceptional work.

Thank you for the constructive and supportive comments on our manuscript.

Referee #2 (Remarks to the Author):

In this manuscript, Shah and colleagues address the evolution of mitotic mechanisms, and more specifically the choice between open and closed mitosis, focusing on a group of unicellular eukaryotes closely related to metazoans. Both strategies have a wide distribution and are sometimes found in the same species, and the question of the conditions that favor one or the other has long been posed. This manuscript presents the characterization of two species of Ichthyosporidia, representing the two known lineages, Dermocystida and Ichthyosporidia. By searching their genomes for orthologs of genes involved in different aspects of mitosis, the authors identify patterns of gene conservation or loss that suggest differences in the mitotic strategies of the two species. Chromosphaera perkinsii possesses genes

known to encode centriole components that *Spheroforma arctica* does not. This is confirmed by electron microscopy of both species, which reveals the presence of centrioles in the former and a structure reminiscent of the SPB of higher fungi in the latter. These analyses also show that, like higher fungi, *S. arctica* uses closed mitosis. In contrast, *C. perkinsii* displays an open mitosis similar to that observed in animal cells: the nuclear envelope breaks down, the spindle is organized by centrioles, and the chromosomes condense to form an equatorial plate. The properties of the spindle (twisting, scaling of inter-kinetochore distance) are comparable to those of human cells. The species most closely related to the Ichthyosporea, *Corallochytrium limacisporum*, also exhibits open mitosis. The authors conclude that open mitosis was probably present in a common ancestor of Ichthyosporea and Corallochytria, and thus pre-existed metazoans. They also conclude that the evolution of closed mitosis in Ichthyosporea and probably in other phyla as well is linked to the presence of a multinucleate stage in the life cycle.

This study is of great interest for understanding the evolutionary origins of Holozoa, to which metazoans belong. The characterization of mitosis in two species belonging to the two major lineages of Ichthyosporea, and the demonstration of their resemblance to the mitoses of animals on the one hand, and fungi on the other, is an important contribution to understanding this evolutionary history. The work is well executed and overall adequately presented. Furthermore, this work brings considerable weight to the hypothesis of a link between closed mitosis and coenocytic development. However, some points need to be improved before it can be considered for publication in Nature.

Thank you, we appreciate your interest in our work and its significance for the field.

First, the authors state that their “analyses produce several experimentally verifiable predictions, including the presence of centrioles in Dermocystida and acentriolar MTOCs in Ichthyophonida by an overall divergence in spindle architecture, NE and NPC dynamics, and kinetochore organisation between the two lineages”. If there are indeed differences in the conservation of some of the genes involved in these processes, it is not obvious how they are predictive, to the exception of the loss of centriole components whose signature is well characterized. For example, lamins are lost in Ichthyophonida, but also in *C. limacisporum*, which nevertheless uses open mitosis. Similarly, it is mentioned that only a truncated form of ELYS is present in *S. arctica*, but no ortholog is identified in *C. limacisporum*. Some of the CenpHIKMs are lost in Ichthyophonida (and not always the same ones), but they are all lost in *C. limacisporum*. Among the Ichthyophonida, *Ichthyophonus hofleri* has virtually all augmin subunits.

We agree with this important and very helpful piece of feedback. The phylogenetic analyses clearly suggest differences in the MTOC and spindle (e.g. Augmin, PRC1) between *S. arctica* and *C. perkinsii*, but indeed, any observed phylogenetic divergences for components of the NE, NPC or kinetochore, although intriguing, are much more difficult to interpret. These differences are even less clear-cut once we take the other holozoan species into account.

Therefore, we have removed any claims of an ability to predict open or closed mitosis or indeed mitotic mechanism per se, and have instead highlighted that the comparative genomics is sufficient to support

the hypothesis that mitotic mechanisms might vary widely between the ichthyosporean species and outgroups analysed here - and to motivate the experimental analyses that must follow.

Additionally, we redid the phylogenetic analysis for the lamin family using a search strategy optimised for a larger eukaryotic search set based on a previously published study into lamin evolution (Koreny et al. GBE, 2016). This led us to confirm the presence of lamin homologs in all Ichthyosporea, including two in *S. arctica*, where previously we had found none.

The relevant sections of the text are copied here for clarity:

Line 114: Although collectively these differences are a strong indication of a divergence in mitotic mode between C. perkinsii and S. arctica, they are not sufficient to determine either species' position along the spectrum from open to closed. Furthermore, inspection of the phylogenetic profiles of the other Ichthyosporea and assorted Holozoa (the group containing animals and their closest relatives) (Extended Data Fig. 1) reveals a significant degree of compositional variability in the complement of mitosis-associated protein complexes, making it difficult to draw any generalisable conclusions in the absence of an accompanying experimental investigation.

Apart from centrioles, the clearest variation is the loss of CenO-P-Q-R/Nkp1-U-Nkp2 - although Nkp1 and Nkp2 are also lost in *C. limacisporum* - and CenpN, but it's not clear how this can be linked to the transition from open to closed mitosis. Other elements, such as the low chromosome condensation observed during mitosis in Ichthyophonida, could possibly also explain some of these variations.

This is correct. We have been careful to avoid any suggestion in the revised text that the kinetochore differences, albeit quite intriguing, predict mitotic mode in any way.

The fact that most of the variations in the genes highlighted by the authors do not point clearly or very convincingly to a change in mitotic strategy is probably due in part to the choice made in presenting the results. Another option would be to start from the observation that the two lineages use different mitotic strategies to identify genes that are lost or modified. Alternatively, the absence of centriolar genes, which is associated in other eukaryotes with the development of divergent MTOCs, would in itself justify a more detailed analysis of mitotic mechanisms. Once the difference between species in terms of mitotic strategies has been confirmed, further genome analysis can be carried out to determine how this affects the conservation of spindle components, the nuclear envelope, etc.

Indeed. Throughout our revised manuscript we present the message as follows: comparative genomics can be used to generate hypotheses about divergent cellular machinery, but follow-up experiments are absolutely essential to identify and characterise phenotypes. As the reviewer suggests, genome comparisons can enable more specific hypotheses once such phenotypic data exists.

The relevant sections of the text are copied here for clarity:

Line 47: *For example, building an intranuclear spindle in closed mitosis must be accompanied by NE fenestration to allow insertion of the MTOC¹⁹. On the other hand, open mitosis requires distinct interphase and post-mitotic mechanisms for the insertion of new nuclear pore complexes (NPCs) into the NE²⁰. These significant differences in the core division machinery imply that certain biophysical constraints linked to the mode of mitosis may be encoded in the genome, enabling the use of comparative genomics to identify new cases of probable divergence between related species outside traditional model systems. We can then combine phylogenetic inference with the targeted experimental investigation of mitotic dynamics in these lineages to ask whether constraints imposed by ecological niche and life cycle could drive species towards either open or closed mitosis.*

Line 284: *Beyond mitosis, our work highlights that genotype alone, although a powerful hypothesis-generator, is insufficient to predict cellular phenotypes that are invariably constrained by ecological niche and life cycles. However, we have access to many more high-quality genomes than experimental model systems, and developing a new species into a model system demands many years of dedicated effort. Here we address that issue using a volumetric ultrastructure imaging approach that combines the scalable tools of expansion microscopy with the traditional advantages of EM. When integrated with phylogenetics, such a framework can enable a comparative approach to investigating diversity and evolution in cell biology.*

- *C. limacisporum* also appears to have a coenocytic stage (doi.org/10.1016/j.cub.2021.06.061), as do other Dermatocystida. Is mitosis closed, and what do MTOCs look like in these species? A life cycle comprising both a uninucleate and a multinucleate stage, with open or closed mitosis, respectively, could have pre-existed the divergence between Corallochytreia and Ichthyosporia. In this case, MTOCs resembling the *S. arctica* MTOC could also be present in Dermatocystida and in *C. limacisporum* during multinucleate stages. Open mitosis would have been lost in the Ichthyosporia - or only in some of them, since some of the gene losses observed in *S. arctica* are not seen in *Amoebidium parasiticum* and *Ichthyosporus hofleri*. The protocol for expansion microscopy combining tubulin and NPC labelling could help to better characterize these transitions.

This is an important comment, and we agree that the physiology of the outgroup to the Ichthyosporia, *C. limacisporum*, has key implications for our model of mitotic evolution. For this revision, we carried out significant additional analyses of the *C. limacisporum* (Clim) life cycle. These are now included in Figure 1, Figure 4, Extended Data Figure 8, and Extended Data Figure 9.

Response Fig.1 (Extended Data Fig. 8). Mitotic strategies in *A. appalachense* and corallochytrean *C. limacisporum*. **b**, Maximum intensity projections of mitosis in *A. appalachense* cells labelled for DNA (grey) and cellular membranes (magenta). **c**, Maximum intensity projections of *C. limacisporum* nuclei through mitosis, labelled microtubules, DNA and NPCs. **d**, *A. appalachense* has two distinct acentriolar MTOCs, cytoplasmic (yellow asterisk) and nuclear-associated (yellow circle). Single slice from TEM tomography of *A. appalachense* interphase nucleus **e**, *C. limacisporum* has a centriolar MTOC. Single slices from TEM tomography of *C. limacisporum* cells showing centrioles (black arrows). Scale bars, 5 μm (**a**), 2 μm (**b** and **c**), 500nm (**d** & **e**).

Unlike the coenocytic stages of species across the tree, including *S. arctica*, *D. melanogaster*, or *P. polycephalum*, Clim coenocytes are present at very low frequency (1-3% depending on growth conditions, this work and consistent with those reported in Kozyzkowska et al. 2021, <https://doi.org/10.1016/j.cub.2021.06.061>). We were not able to increase the frequency of coenocyte formation through temperature stress beyond 3%, and mitotic coenocytes are vanishingly rare (we were able to find and image only 2 examples). We have never observed multinucleated cells with more than 6 nuclei, and there appears to be no mechanism for spacing nuclei or regulating nuclear positioning in any way (Extended Data Fig. 9b and c), suggesting an upper limit on the capacity of cells to sustain

this state. We also observed cells with 3, 5 or 6 nuclei (Extended Data Fig 9c), suggesting that mitosis is sometimes asynchronous, reflecting either a lack of tight regulation or a certain rate of mitotic failure in the coenocytes. Taking these considerations together, we did not expect to find significant rewiring of mitosis between states in this system, if indeed the coenocytic stage represents a stable state, in sharp contrast to, for example, *Physarum*.

Consistent with our expectations, we show that Clim has centrioles in uninucleate, binucleate and multinucleate states (Extended Data Fig. 8e and 9d) with an open mitosis (loss of NPCs and apparent disruption of the NE barrier as observed by NHS ester staining) indistinguishable between uninucleate and multinucleate cells (Fig. 4 and Extended Data Fig. 9e). In contrast to *C. perkinsii* cells forced into a multinucleated state by induced cytokinesis failure (Extended Data Figure 9g-i), which exhibit multipolar spindle formation and catastrophic mitotic failure, the two examples of mitotic Clim coenocytes we were able to observe possessed normal bipolar spindles. This suggests that Clim is able to prevent pole fusion in some way, enabling the survival of the coenocytic cells. Perhaps this is through partial retention of the NE barrier, as we observed by TEM tomography (Extended Data Figure 9f). Taking our findings together, we would suggest that *C. limacisporum* carries out an open mitosis while retaining some control over NE barrier function.

Line 238: *In contrast, and consistent with the phylogenetic analysis (Fig. 1a and Extended Data Fig. 1), the predominantly uninucleate life cycle of the outgroup corallochytrean C. limacisporum⁴⁰ appears to be facilitated by open mitosis that relies on a spindle nucleated from centriolar MTOCs (Fig. 4a and Extended Data Fig. 8c and e). Although C. limacisporum proliferates through binary fission, with a characteristic long gap between mitosis and cytokinesis that results in a large proportion of binucleated cells at steady state (Fig. 1a and Extended Data Fig. 8), a small proportion of cells have been reported to exist as coenocytes⁴⁰. These coenocytes, representing at most 3% of the population, divide rarely and often asynchronously, reaching a maximum of 8 irregularly spaced nuclei (Extended Data Fig. 9a-c)⁴⁰. Although we observed no difference between the open mitosis of the uninucleated cells and that of the coenocytes, a limited capacity for coenocytic division in this species might be supported by asynchronous mitoses and partial remnants of the NE barrier observable by ET (Extended Data Fig. e and f). In contrast, C. perkinsii cells forced to enter a multinucleated state through the induction of cytokinetic failure, exhibit multipolar spindles and major mitotic defects (Extended Data Fig. 9 g-i), much like animal cells^{41,42}, highlighting the incompatibility of fully open mitosis with coenocytic divisions.*

Zooming out, the implications for holozoan mitotic evolution are as follows: the holozoan ancestor likely had either a) a life cycle comprising both open and closed mitosis linked to distinct uninucleated and multinucleated stages, like *Physarum*, or b) an intermediate form of open mitosis with some barrier function, like Clim or *Drosophila*. This was then followed by reductive evolution towards either the open or closed state, as in animals, fungi, choanoflagellates, and the various Ichthyosporea, coupled to a predominance of a simplified uninucleated or multinucleated life cycle respectively.

Specifically for the Ichthyosporea, our additional data on *Amoebidium*, *C. fragrantissima* and additional *Sphaeroforma* species added in revision (Figure 1, Figure 4, Extended Data Figure 8) suggests that the coenocytic state coupled to closed mitosis is a defining phenotypic feature of the Ichthyophonida.

That said, there does not seem to be any consensus as to the greater phylogenetic proximity between *C. limacisporum* and the Ichthyosporea within the Holozoa (see doi.org/10.1016/j.cub.2017.06.006). What are the arguments in favor of the phylogeny presented here? It could indeed change the interpretation of the observations.

As to the phylogenetic position of *C. limacisporum*, we have not constructed any species trees for this work, relying instead on recently published work that represents the state of the art (Ocaña-Pallarès et al. 2023, <https://www.nature.com/articles/s41586-022-05110-4>).

Other comments :

- When does the duplication of *S. arctica* MTOC occur? Have the authors observed any duplication intermediates?

To address this question, we induced S-phase arrest in *S. arctica* cells using hydroxyurea (HU), which results. Cells treated with HU exhibited duplicated MTOCs with near-complete penetrance of the phenotype, indicating that MTOC duplication likely occurs during the G1/S transition and proceeds independently of DNA replication (Extended Data Fig. 3a and 3b).

- It is interesting to find some centriolar proteins (SAS6, CEP135, Plk4) but not all (CPAP, TUBE, TUBD) in *Amoebidium parasiticum* and *Ichthyophonus hofleri*. Could there be a simplified centriole in these species like for instance in *C. elegans*? And what about their life cycles: are they only dividing as coenocytes? Here again, expansion microscopy could help address these questions relatively easily.

The *Amoebidium parasiticum* genome has a high level of fragmentation and may lead to some proteins, especially small ones, not being detected. To this end, we analysed a new better-quality genome of *Amoebidium appalachense* belonging to the same genus (<https://doi.org/10.1101/2024.01.08.574619>). Here we were able to now additionally detect CPAP while TubD & TubE remain absent, with high confidence given the deep conservation of tubulin proteins across eukaryotes.

It is indeed of great interest that *Amoebidium* species encode some centriolar proteins, and we examined this in depth. The coenocytic stage is the only stage we observe in *A. appalachense* under our lab culture conditions. While amoebae have been reported previously in the presence of dialysed fish extracts ([https://doi.org/10.1016/0012-1606\(68\)90005-5](https://doi.org/10.1016/0012-1606(68)90005-5)), the number of nuclei in these amoebae and whether they undergo mitosis remain unknown. We studied coenocytic *A. appalachense* using U-ExM, TEM tomography, and FIB-SEM. No centrioles were detected, but we were able to identify two distinct types of acentriolar MTOCs: one cytoplasmic and another nuclear-associated (Extended Data Fig. 8d).

Moreover, using U-ExM (Fig. 4a) which only worked following high-pressure freezing (Cryo-ExM) (<https://doi.org/10.1242/jcs.260240>) in this system, followed by pan-labelling of the proteome with NHS-ester combined with BODIPY lipid staining (Extended Data Fig. 8b), we reveal that *A. appalachense* carries out a closed mitosis. Together, all these results are consistent with the general framework we

propose in this work, in which coenocytic ichthyophonids depend on an acentriolar MTOC and undergo closed mitosis.

- The image of *C. limacisporum* (Extended data Fig.8) is not very informative as such. An image of an interphase cell showing the NPCs should be added for comparison. Also, the image of *C. fragrantissima* is of poor quality, and the NPCs are not visible at all.

To answer this comment, we now provide better images for both *C. fragrantissima* and *C. limacisporum* in Fig.4a and Extended Data Fig. 8c). Our new data provides clearer evidence that *C. fragrantissima* undergoes closed mitosis whereas *C. limacisporum* undergoes open mitosis.

- Members of the Augmin complex are referred to as HAUS in Extended data Fig.1
In the figure, we have updated the designation of HAUS proteins as members of the Augmin complex.

-l101: "centromere associated network (CCAN) network".

This has been corrected.

Referee #3 (Remarks to the Author):

(A, B, C, E, H) In this manuscript, the authors have investigated the manner of mitosis in the clade Ichthyosporia, close relatives of animals. Dey and his co-workers have been studying cells of eukaryotes in order to know how (and why) they evolved two extreme strategies of cell division, namely, open and closed mitosis. They have recently found that fission yeast uses a surprisingly similar mechanism of nuclear remodeling to animals, even though it adopts closed mitosis strategy, in contrast to animals which mostly uses the open strategy. Their achievement has been outstanding, as they not only found a common molecular mechanism between the two different mitotic strategies, but also elucidated how eukaryotes keep the integrity of nuclear envelope during closed mitosis, in which the membrane would be highly expanded.

Now, they focus more on the divergence of the strategies, using Ichthyosporia which contains two closely-related clades that undergo open or closed mitosis. They have found that *S. arctica*, which proliferates as a coenocyte, adopts the closed strategy, using state of the art techniques including TEM tomography, the focused ion beam scanning electron microscopy, and antibody staining based on the ultrastructure expansion microscopy. In contrast, another ichthyosporian *C. perkinsii* undergoes open mitosis, in a surprisingly similar manner as humans. They have concluded that the momentum that drove Ichthyosporia towards closed mitosis would be the restriction from their life cycle, namely, the multi-nucleated development, which may require an additional setup to ensure the faithful segregation in a shared space. Their findings have strongly reinforced the previous notions obtained from *Drosophila* embryo, animal germ lines, hyphal growth of fungi, and some amoebozoans.

Their conclusion is supported by a line of solid evidence that were produced by amazingly well-developed microscopic techniques. Their work has provided an explanation how ichthyosporians have adopted their strategies of mitosis along with the restriction from life cycle, filling the gap between previous studies on fungi and animals. However, I do not think this manuscript represents the greatest scientific

significance, compared to their previous work on fission yeast. I recommend authors to publish it in a more specialized journal.

Thank you for your appreciation of our work and for recognizing its significance in furthering our understanding of mitotic evolution.

(D) [Statistics] Statistics are correctly used.

(F) [Improvement] None specifically. However, I feel that they need to experimentally prove their conclusion that the nuclear envelope of multi-nucleated organisms is indispensable for faithful segregation of nuclear materials using, for example, genetically modified *S. arctica* (or other Ichthyophonida species) whose nuclear envelope are disturbed.

Currently, there are no genetic tools for *S. arctica* or other ichthyophonids, nor do chemical methods exist to perturb NE integrity using small molecule inhibitors. Therefore, we undertook an alternative approach to address this question. We used blebbistatin, a Myosin II inhibitor used widely across eukaryotic systems, on *C. perkinsii* to block cytokinesis and induce the formation of multinucleated cells in the context of an open mitosis. As detailed in Extended Data Fig. 9, our findings reveal that multinucleated *C. perkinsii* cells experience a marked increase in mitotic defects, characterised by multipolar spindles and fragmented nuclei. This phenotype, also observed in various animal systems (Gönczy, P. *et al.*, 2000, Goupil, A. *et al.*, 2020), suggests that multinucleated cells require mechanisms to protect against the mixing of nuclear contents in a shared cytoplasm.

(G) [Reference] I don't find any missing references. However, the authors need to clarify from where the *S. arctica* and *C. perkinsii* were obtained.

The source of species is now highlighted in greater detail in both the method section and acknowledgements.

Reviewer Reports on the First Revision:

Referees' comments:

Referee #1 (Remarks to the Author):

The revised manuscript has been significantly improved, both in terms of content and presentation. The authors have carefully considered and addressed all the advice, questions, and criticisms from the reviewers in detail. The message of this publication has become even clearer. I believe it will find its way into many student lectures. Although one could argue that 'general relevance' could only be achieved through the study of mainstream models, I consider this view outdated, especially in light of the vast range of cell biological biodiversity. As August Krogh formulated it more than 100 years ago, "For many problems there is an animal or plant on which it can be most conveniently studied." In the present paper, perhaps, these are the Ichthyosporea. This work deserves to be published in Nature.

Referee #2 (Remarks to the Author):

The authors have included important new results and have responded well to all the questions that were raised. I therefore now fully support the publication of this work.

Minor comment: You should indicate the nuclear envelope remnants in *C. Limacisporum* mitotic cells for clarity. In addition, there are some small errors in the indications on the supplementary figures (purple arrows and not black, for example).

Referee #3 (Remarks to the Author):

In the previous comment, I suggested the authors more specialised journals for publishing their excellent results. I had in my mind Nature Cell Biology or even Cell, which would be more appropriate for the topic. After the revise, the story becomes simpler and perhaps more interesting to general readers. However, it still seems a bit less appealing before publishing in Nature. Thus I would like to suggest the following two points.

1. Better evolutionary explanation and discussion

First of all, please add some explanation (e.g. a few sentence in the introduction) why evolution of closed and open mitosis is important for general biology. Difference of these two types of mitosis does not seem to represent a very important evolutionary event compared to the other "big" evolution such as the evolution of eukaryote by acquiring nuclear membrane, or the evolution of mitochondria by symbiogenesis. One of the reasons is that the invention of open or closed mitosis does not seem to trigger evolution of significant biological traits (such as the increased genetic materials or the greater energy production by aerobic respiration) in the history of life. In other words, the evolution of open/closed mitosis seems just a consequence of another evolutionary event, which may be the evolution of coenocyte as the authors stated. Adding some sentences justifying this point would greatly appeals to general readers.

Second, please explain more clearly your hypothesis on the evolution of open/closed mitosis in the course of eukaryote evolution. By textbook knowledge, it is generally considered that the Eukaryota started with closed mitosis, and then several lineages secondary evolved open mitosis. If it still represents the general consensus (which is not very clear in the manuscript), then the authors does not need to postulate a holozoan ancestor that is capable of open mitosis in addition to closed mitosis; it may be equally possible that several lineages of the Opisthokonta independently ended up with open mitosis, because they did not need to maintain the closed mitosis anymore by somereasons. Similarly, the authors' last corollary "once closed mitosis has evolved, it can persist even if the organism returns to a unicellular, uninucleate life cycle" is confusing, because it sounds like being based on an assumption that open mitosis was the ancestral state in the Eukaryota. Please consider rephrasing it.

2. More direct evidence using genetic tools, even a transient one

In the manuscript, whether *S. arctica* or other ichthyophonids undergo closed mitosis obviously represents the most critical point of the story. However, the evidence is nearly exclusively based on microscopic observations, which unfortunately reduces the strength of the authors' claim. More direct evidence would be demanded for convincing general readers. The authors executed a clever alternative using blebbistatin, but it is a too strong treatment to say something reasonable. What about a more gentle experiment? The authors wrote in the reply "currently, there are no genetic tools for *S. arctica* or other ichthyophonids". I do not think it is true. Unless I am terribly wrong, *Creolimax* is genetically tractable (or may be other species as well), even though the effect is transient. By knocking down the genes that are important on constructing NE, or over-expressing such genes with critical mutation, the authors could mildly disturb the formation of NE. Even a rather simple experiment expressing GFP with nuclear localisation signal (NLS) would prove if the NE is intact during mitosis (GFP will mostly be kept in the nuclei during M phase). Or more simply put, the NE can be visualised in live cells by expressing a GFP-labelled NE-embedded protein to see if it does not collapse during mitosis.

Author Rebuttals to First Revision:

RESPONSE TO REFEREES

We sincerely appreciate your time, effort and encouraging feedback. It has been our main priority to generate the data and improve the text to best address your comments and concerns. The revised paper is, in our opinion, far stronger as a result of your constructive and insightful comments. Thank you.

Throughout this response document, we use **blue text** to indicate our response, **black text** to indicate the referee's original comment, and **magenta text** to indicate a quote taken from the manuscript.

In the manuscript, changes in the text are highlighted in **yellow**.

Referees' comments:

Referee #1 (Remarks to the Author):

The revised manuscript has been significantly improved, both in terms of content and presentation. The authors have carefully considered and addressed all the advice, questions, and criticisms from the reviewers in detail. The message of this publication has become even clearer. I believe it will find its way into many student lectures. Although one could argue that 'general relevance' could only be achieved through the study of mainstream models, I consider this view outdated, especially in light of the vast range of cell biological biodiversity. As August Krogh formulated it more than 100 years ago, "For many problems there is an animal or plant on which it can be most conveniently studied." In the present paper, perhaps, these are the Ichthyosporia. This work deserves to be published in Nature.

Thank you very much for the generous and encouraging feedback, and message of support for our work and also for the broader field in general - we greatly appreciate it.

Referee #2 (Remarks to the Author):

The authors have included important new results and have responded well to all the questions that were raised. I therefore now fully support the publication of this work.

Minor comment: You should indicate the nuclear envelope remnants in *C. Limacisporum* mitotic cells for clarity. In addition, there are some small errors in the indications on the supplementary figures (purple arrows and not black, for example).

Thank you for fully endorsing our revised manuscript - your feedback was valuable in strengthening the paper and sharpening its interpretations and conclusions.

We have made the requested figure edits. NE remnants in *C. limacisporum* mitotic nucleus are now marked in blue (Extended Data Fig. 10f). Thank you for spotting the error with the colours of the arrows. The legend is corrected to match the colour of the arrows. (Extended Data Fig. 9e

and 10f).

Referee #3 (Remarks to the Author):

In the previous comment, I suggested the authors more specialised journals for publishing their excellent results. I had in my mind Nature Cell Biology or even Cell, which would be more appropriate for the topic. After the revise, the story becomes simpler and perhaps more interesting to general readers. However, it still seems a bit less appealing before publishing in Nature. Thus I would like to suggest the following two points.

Thank you for your feedback and for endorsing the revised version of our manuscript; we are glad that you continue to find the work of general interest. We have added significant new data and modified the text in order to address your suggestions for improving the manuscript further. Please find our detailed responses to each point below.

1. Better evolutionary explanation and discussion

First of all, please add some explanation (e.g. a few sentence in the introduction) why evolution of closed and open mitosis is important for general biology. Difference of these two types of mitosis does not seem to represent a very important evolutionary event compared to the other “big” evolution such as the evolution of eukaryote by acquiring nuclear membrane, or the evolution of mitochondria by symbiogenesis. One of the reasons is that the invention of open or closed mitosis does not seem to trigger evolution of significant biological traits (such as the increased genetic materials or the greater energy production by aerobic respiration) in the history of life. In other words, the evolution of open/closed mitosis seems just a consequence of another evolutionary event, which may be the evolution of coenocyte as the authors stated. Adding some sentences justifying this point would greatly appeals to general readers.

This is a useful piece of feedback and we have made small changes to the introduction to more clearly emphasise that evolution towards a more open or closed mitosis, events that have clearly occurred multiple times in eukaryotic evolution, would be tightly coupled to other life cycle or ecological constraints rather than act as a major driving force for evolutionary innovation in isolation. Open and closed mitosis might each provide specific and distinct benefits to the cell: for example, closed mitosis could prevent mixing of nuclear contents, the central thesis of this work, and open mitosis could aid in cellular rejuvenation through complete recycling of most of the nuclear proteome once every cell cycle. In our paper, as you state in your review, we provide evidence for the former, but we prefer not to speculate on the benefits of open mitosis at this time as we have not directly addressed this experimentally in any way.

Line 40-45: Although open and closed mitosis have each likely evolved independently multiple times in different branches of the eukaryotic tree^{9,10}, with many unique lineage-specific adaptations resulting in a broad distribution of intermediates from fully open to fully closed^{1,2}, the evolutionary pressures that drive species towards the extremes of either mitotic strategy are not well understood.

Second, please explain more clearly your hypothesis on the evolution of open/closed mitosis in the course of eukaryote evolution. By textbook knowledge, it is generally considered that the Eukaryota started with closed mitosis, and then several lineages secondary evolved open mitosis. If it still represents the general consensus (which is not very clear in the manuscript), then the authors does not need to postulate a holozoan ancestor that is capable of open mitosis in addition to closed mitosis; it may be equally possible that several lineages of the Opisthokonta independently ended up with open mitosis, because they did not need to maintain the closed mitosis anymore by some reasons. Similarly, the authors' last corollary "once closed mitosis has evolved, it can persist even if the organism returns to a unicellular, uninucleate life cycle" is confusing, because it sounds like being based on an assumption that open mitosis was the ancestral state in the Eukaryota. Please consider rephrasing it.

This is a helpful suggestion. We agree that we could provide more clarity on these points, and have edited the text accordingly.

It has indeed been assumed for a long time that the ancestral mode of mitosis in the last common eukaryotic ancestor (LECA) was closed (Heath 1980, Sazer 2014). However, there is no direct evidence either strongly supporting or disproving this model - acquiring such evidence requires careful experimental study with modern tools of deep-branching lineages across the eukaryotic tree, most of which are small heterotrophic flagellates thought to represent an approximation of what the LECA cell might have looked like (Derelle 2015, Tromer 2019, Azimzadeh 2021). Similarly, the mitotic phenotype of the opisthokont ancestor (the Holozoa plus the Holomycota) is also not predicted with any confidence, because reconstructing this would require a full comparative analysis of holozoans, fungi and their close relatives, as well as the apusomonad and breviate outgroups (Dresler & McAinsh 2012, Burki 2019). Therefore we restrict our discussion to models dealing with the holozoan ancestor and nothing older.

In our work, we show that both open and closed mitosis are distributed across the Ichthyosporea with indications of a semi-open mitosis in the Corallochytria, the immediate outgroup. This suggests several plausible scenarios for the holozoan ancestor, including fully open, fully closed, or an intermediate form, possibly accompanied by a complex life cycle that alternated between these states depending on whether it was multinucleated or uninucleated. We have now clarified the text to communicate clearly that we cannot at this point distinguish between these different models.

Our comment about closed mitosis persisting even if the organism returns to a unicellular, uninucleate life cycle has specifically to do with the evolution of unicellular yeasts from multinucleate hyphal ancestors (Gladfelter 2006, Baidouri et al 2021), not deeper branches of the tree. We have now clarified this to prevent any possible confusion.

Line 277: The centriole-dependent open mitosis of *C. limacisporum* (Fig. 4 and Extended Data Fig. 9 and 10) together with its limited capacity for coenocyte formation suggests several plausible scenarios for mitosis in the holozoan ancestor, including open, closed, or an intermediate form, possibly accompanied by a complex life cycle that alternated between these

states depending on whether it was multinucleated or uninucleate. Such a life cycle would be closely analogous to that of *Physarum polycephalum* (Fig. 4), which transitions from a crawling uninucleate amoeba with centrioles and open mitosis to a giant acentriolar coenocyte carrying out a closed mitosis⁴⁴. From this putative ancestor, the dermocystids inherited a flagellated life cycle stage^{28,40} and, as we show in this study for *C. perkinsii*, specialised towards palintomic divisions through open mitosis (Fig. 3 and Extended Data Fig. 8); features shared with the ancestor of animals. In contrast, the adoption of an exclusively coenocytic life cycle within the Ichthyosporaea was accompanied by the loss of the centriole, the *de novo* evolution or retention of a distinct, nucleus-associated MTOC, and a fully closed mitosis.

Line 299: A corollary of our hypothesis is that closed mitosis can persist even when the organism evolves a unicellular, uninucleate life cycle^{7,56}, as in yeasts evolving from hyphal fungal ancestors, but in such cases is apparently no longer under strict selection to remain closed^{57,58}.

2. More direct evidence using genetic tools, even a transient one

In the manuscript, whether *S. arctica* or other ichthyophonids undergo closed mitosis obviously represents the most critical point of the story. However, the evidence is nearly exclusively based on microscopic observations, which unfortunately reduces the strength of the authors' claim. More direct evidence would be demanded for convincing general readers. The authors executed a clever alternative using blebbistatin, but it is a too strong treatment to say something reasonable.

We understand your emphasis on having a weight of evidence in favour of the conclusion that *S. arctica* (and other ichthyophonids) undergoes closed mitosis. We believe that our study already provides this evidence from multiple sources, to which we now add several pieces of data that further strengthen our argument.

1. We have included more time series of live imaging data using the membrane dye FM 4-64 in *S. arctica* (previously only supplementary movie 2, now also in Extended Data Fig. 4). These light sheet movies, acquired at 1 minute intervals, show that FM4-64 is completely excluded from the nuclear volume (see also Olivetta 2023) throughout the mitotic process, without any apparent disruption to NE integrity. Furthermore, these movies allow us to characterise the series of characteristic shape changes (indicative of a closed mitosis) accompanying nuclear division in live cells.

2. In previous versions of the manuscript, we already demonstrated that nuclear pore complexes (NPCs) remain intact throughout the mitotic process, with no reduction in NPC density during nuclear division (Figure 2c and f). Since NPCs can only be integrated into a fully intact double-layered NE, the retention of NPCs provides additional certainty that NE integrity is maintained through mitosis.

3. We have included data on fixed cells, imaged with expansion microscopy and labelled using lipid dye BODIPY ceramide. BODIPY labels the NE with high contrast as a continuous contour

throughout the nuclear cycle, providing yet another line of evidence to support the conclusion that the NE remains intact throughout nuclear division. These data can now be found in Figure 2e.

4. Finally, as transmission electron microscopy (TEM) tomograms are considered the gold standard, across model systems, for demonstrating the structural integrity of the NE at the nanoscale (see for example Dey et al. Nature 2020), we have included an additional tomogram of cells in early mitosis (previously only supplementary movie 5, now also in Figure 2g top) to complement the tomogram of an anaphase nucleus in Figure 2. These tomograms showcase, once again, with no ambiguity, an intact NE during mitosis.

What about a more gentle experiment? The authors wrote in the reply “currently, there are no genetic tools for *S. arctica* or other ichthyophonids”. I do not think it is true. Unless I am terribly wrong, *Creolimax* is genetically tractable (or may be other species as well), even though the effect is transient.

The authors of this study (first the Dudin lab and subsequently the Dey lab) have been attempting to transiently transform *Sphaeroforma arctica*, the primary ichthyophonid model in our study, for more than five years (dx.doi.org/10.17504/protocols.io.z6ef9be) without success.

We thus assume here that you are referring to the study by Suga et al. (2013), which is the sole publication demonstrating a transient transformation of an ichthyophonid, specifically *Creolimax fragrantissima*. Despite our efforts to replicate these findings in one of our laboratories (Dudin lab), we have been unsuccessful for several years. It is important to note that, since 2013, no other laboratories, including the original authors, have successfully applied these methods for genetic manipulation or live cell imaging of any process. We attribute these challenges to the low efficiency of electroporation, the lack of a successful selection strategy (dx.doi.org/10.17504/protocols.io.z5nf85e), and the difficulties in handling cells, since detachment from the flask (e.g. to enrich using a cell sorter) is highly toxic, complicating the maintenance of cultures over time for any live cell imaging.

Although developing genetic tools for our ichthyosporean models remains a critical goal for the future that will likely require several more years of concerted effort from our research groups, it lies outside the scope of our current study.

By knocking down the genes that are important on constructing NE, or over-expressing such genes with critical mutation, the authors could mildly disturb the formation of NE.

Even if genetic manipulation were possible in *S. arctica*, most perturbations to NE function in better-studied human and yeast models lead to dominant interphase phenotypes related to other key roles of the NE in genome organisation, transcription and regulation of transport (for a recent review: <https://febs.onlinelibrary.wiley.com/doi/full/10.1002/1873-3468.14769>), and do not enable us to learn much about the role of the NE in mitosis.

However, there is a notable exception - closed mitosis in the fission yeast *S. pombe* requires a fairly dramatic expansion of NE surface area, through the delivery of newly synthesised phospholipids, to enable shape changes in anaphase (Yam et al. Current Biology 2011, Dey et al. Nature 2020). This can be blocked using the drug cerulenin, which leads to spindle buckling, asymmetric NE partitioning, and in some cells complete failure of nuclear division. One would not expect such dramatic phenotypes in an open mitosis - for example, no NE expansion was observed in the related fission yeast *S. japonicus* that undergoes a semi-open mitosis through anaphase tearing of the NE (Yam et al. Current Biology 2011).

We treated *S. arctica* with cerulenin, which leads to an arrest of cell growth and produces nuclei with 4 spindle poles as a consequence of a failed round of nuclear division - consistent with the data from *S. pombe*. This data is now in Extended Data Figure 6a and b. We believe that this provides additional functional evidence for a canonical closed mitosis in *S. arctica*.

Even a rather simple experiment expressing GFP with nuclear localisation signal (NLS) would prove if the NE is intact during mitosis (GFP will mostly be kept in the nuclei during M phase). Or more simply put, the NE can be visualised in live cells by expressing a GFP-labelled NE-embedded protein to see if it does not collapse during mitosis.

Even if it were possible to introduce an NLS-GFP construct, changes in NLS-GFP levels are difficult to interpret because they report both on NE integrity as well as changes in NPC transport kinetics. It is a well-documented phenomenon of some closed mitoses that import and export rates are altered during mitosis without loss of NE integrity (DeSouza & Osmani, 2007), and human cells in interphase can exhibit altered rates of nucleocytoplasmic transport without loss of NE integrity (Andreu et al., 2022). Even if such a change occurs in *S. arctica*, although it would be of general interest, it would have no impact on the conclusions of our manuscript, and therefore we focus instead on your suggestion to visualise the NE using additional markers.

In lieu of a GFP-tagged NE protein, since we are unable to transform the cells, we provide time-lapse imaging in live cells using FM4-64 as well as BODIPY labelling in fixed cells with expansion microscopy - each of which reports on the contour of the nuclear envelope throughout mitosis. This data is now in Figure 2e and Extended Data Fig. 1. Consistent with the other data provided in the manuscript, these experiments all indicate an intact NE throughout mitosis with no detectable loss of integrity.

References

Heath, I. B. Variant Mitoses in Lower Eukaryotes: Indicators of the Evolution of Mitosis? in *International Review of Cytology* vol. 64 1–80 (Elsevier, 1980).

Sazer, S., Lynch, M. & Needleman, D. Deciphering the Evolutionary History of Open and Closed Mitosis. *Current Biology* 24, R1099–R1103 (2014).

Derelle, R., Torruella, G., Klimeš, V., Brinkmann, H., Kim, E., Vlček, C., Lang, B. F. & Eliáš, M. Bacterial proteins pinpoint a single eukaryotic root. *PNAS* 112 (7) E693-E699 (2015).

Tromer, E. C., van Hooff, J. J. E., Kops, G. J. P. L. & Snel, B. Mosaic origin of the eukaryotic kinetochore. *Proc Natl Acad Sci USA* 116, 12873–12882 (2019).

Azimzadeh, J. Evolution of the centrosome, from the periphery to the center. *Current Opinion in Structural Biology* 66, 96-103 (2021).

Drechsler, H. & McAinsh, A. D. Exotic mitotic mechanisms. *Open Biol.* 2, 120140 (2012).

Burki, F., Roger, A.J., Brown, M.W., Simpson, A.G.B. The New Tree of Eukaryotes. *Trends Ecol Evol.* 35(1), 43-55 (2019).

Gladfelter, A.S. Nuclear anarchy: asynchronous mitosis in multinucleated fungal hyphae. *Curr Opin Microbiol.* 9(6):547-52 (2006).

El Baidouri F, Zalar P, James TY, Gladfelter AS, Amend AS. Evolution and Physiology of Amphibious Yeasts. *Annu Rev Microbiol.* 8(75), 337-357 (2021).

Olivetta, M. & Dudin, O. The nuclear-to-cytoplasmic ratio drives cellularization in the close animal relative *Sphaeroforma arctica*. *Current Biology* S096098222300307X (2023) doi:10.1016/j.cub.2023.03.019.

Dey, G. & Baum, B. Nuclear envelope remodelling during mitosis. *Curr Opin Cell Biol* 70, 67–74 (2021).

Fahrenkrog, B. & Gasser, S. M. Structure and function of the nuclear envelope and nuclear pores. *FEBS Letters.* 597(22), 2703-2704 (2023)

Yam, C., Gu, Y. & Oliferenko, S. Partitioning and remodeling of the *Schizosaccharomyces japonicus* mitotic nucleus require chromosome tethers. *Curr Biol* 23, 2303–2310 (2013).

De Souza, C.P. & Osmani, S.A. Mitosis, not just open or closed. *Eukaryot Cell.* 6(9), 1521-1527, (2007).

Andreu, I., Granero-Moya, I., Chahare, N.R., Clein, K., Molina-Jordán, M., Beedle, A.E.M., Elosegui-Artola, A., Abenza, J.F., Rossetti, L., Trepát, X., Raveh, B., Roca-Cusachs, P. Mechanical force application to the nucleus regulates nucleocytoplasmic transport. *Nat Cell Biol.* 24(6), 896-905 (2022).

Reviewer Reports on the Second Revision:

Referees' comments:

Referee #3 (Remarks to the Author):

I would like to thank the authors to add more data to increase the interest of general readers, which are not necessarily the specialists of the field. Here are my minor comments.

1. Better evolutionary explanation and discussion

The authors addressed this issue properly. It is now clear that the evolution of closed/open mitosis is rather coupled to life cycles or ecological constraints than some evolutionary innovations. I do also understand that the textbook knowledge that LECA would have undergone closed mitosis is obscure and the evolution of open/closed mitosis is not unidirectional.

Given these, I still have (and the readers would also have) a question: How can eukaryotes go back and forth closed/open mitosis even with the inevitable specialization/adaptation of the core division machinery? If the manner of mitosis is tightly coupled with the life cycles or ecological constraints, eukaryotes need to keep genetic components both for open and for closed mitosis throughout the course of evolution, in which the organisms have to overcome numerous environmental changes. It is true, as the authors stated, that the genetic footprint alone does not provide a solid prediction of closed or open mitosis, but we can neither deny a clear genetic adaptation (as observed in Figure 1) by losing some key components that are necessary for the other style of mitosis. And re-gaining the lost genetic elements is nearly impossible without taking lateral gene transfer into account.

I do understand the authors would like to define the arguments only within the Opisthokonta, but the readers of Nature would be keen to know a general law that can reign the Eukaryota.

Here comes my humble speculation. May the requirement of keeping the genetic component both for open and for closed mitosis during evolution possibly account for the only insufficient prediction provided by the genetic footprint? In other words, a eukaryote cannot perhaps completely discard the genetic components that are necessary for one style of mitosis even if it evolved into the other style, in order to prepare for a somewhat flexible switching of the manner of mitosis. Given that, those that adapted their genetic components completely to one extreme mitotic style may have been extinct upon environmental changes. I am not sure if this makes sense, but I think authors should anyhow solve the inconsistency between the obvious signature of genetic adaptation and the non-unidirectional evolution of mitotic styles.

>A corollary of our hypothesis is that closed mitosis has can persist... (line 299)

The "has" may have to be omitted.

2. More direct evidence using genetic tools, even a transient one

The new imaging data using FM 4-64 has provided a piece of solid evidence that may compensate the lack of genetic manipulation. The readers would be convinced that *S. arctica* undergoes closed mitosis.

> Despite our efforts to replicate these findings in one of our laboratories...

Although what I have actually found is Suga & Miller (2018), where they disturbed the tyrosine

kinase signalling using the transient overexpression, I have understood that the genetic manipulation technique in ichthyosporean is still too immature.

What I am concerned is that the readers may wonder why the authors used *S. arctica*, but not *C. fragrantissima* at all, on which there is an open (even though still narrow) path to directly address the question using the genetic tools. Please add a small clarification that the transformation efficiency is too low (or any other appropriate description) in *C. fragrantissima* to do so, perhaps at the sentence where the authors refer *S. arctica* as the best-studied ichthyosporean model.

Reading through the revised manuscript, I have two more comments.

1. The word “cellularisation” is spelled “cellularization” at line 69.
2. The abbreviation “Cfra” is not necessary in Fig. 1.

Author Rebuttals to Second Revision:

We sincerely appreciate your insightful feedback. We have improved the text to best address your comments and concerns. The revised paper is, in our opinion, much better as a result of your constructive and insightful comments. Thank you.

Throughout this response document, we use **blue text** to indicate our response, **black text** to indicate the referee's original comment, and **magenta text** to indicate a quote taken from the manuscript.

Referees' comments:

Referee #3 (Remarks to the Author):

I would like to thank the authors to add more data to increase the interest of general readers, which are not necessarily the specialists of the field. Here are my minor comments.

We are glad you appreciate the new data included with this revision..

1. Better evolutionary explanation and discussion

The authors addressed this issue properly. It is now clear that the evolution of closed/open mitosis is rather coupled to life cycles or ecological constraints than some evolutionary innovations. I do also understand that the textbook knowledge that LECA would have undergone closed mitosis is obscure and the evolution of open/closed mitosis is not unidirectional.

Given these, I still have (and the readers would also have) a question: How can eukaryotes go back and forth closed/open mitosis even with the inevitable specialization/adaptation of the core division machinery? If the manner of mitosis is tightly coupled with the life cycles or ecological constraints, eukaryotes need to keep genetic components both for open and for closed mitosis throughout the course of evolution, in which the organisms have to overcome numerous environmental changes. It is true, as the authors stated, that the genetic footprint alone does not provide a solid prediction of closed or open mitosis, but we can neither deny a clear genetic adaptation (as observed in Figure 1) by losing some key components that are necessary for the other style of mitosis. And re-gaining the lost genetic elements is nearly impossible without taking lateral gene transfer into account.

I do understand the authors would like to define the arguments only within the Opisthokonta, but the readers of Nature would be keen to know a general law that can reign the Eukaryota.

Here comes my humble speculation. May the requirement of keeping the genetic component both for open and for closed mitosis during evolution possibly account for the only insufficient prediction provided by the genetic footprint? In other words, a eukaryote cannot perhaps completely discard the genetic components that are necessary for one style of mitosis even if it evolved into the other style, in order to prepare for a somewhat flexible switching of the manner of mitosis. Given that, those that adapted their genetic components completely to one extreme mitotic style may have been extinct upon environmental changes. I am not sure if this makes sense, but I think authors should anyhow solve the inconsistency between the obvious signature of genetic adaptation and the non-unidirectional evolution of mitotic styles.

We appreciate this point and indeed are in agreement that the ability to switch between open and closed mitosis on relatively short evolutionary or even developmental timescales implies that both modes of mitosis rely on the same core molecular machinery that can be easily repurposed to support one or the other mechanism. We have edited the discussion to better highlight this perspective.

Line 230-235: The ichthyophonid pathway to specialisation provides an intriguing parallel to events in fungal evolution¹⁸. **Zooming out, the ability of species to adopt divergent mitotic strategies over relatively short evolutionary timescales, or even within the same life cycle as in *Physarum*- (Fig. 4), suggests an explanation for the limited predictive power of our phylogenetic profiles: cells are able to repurpose the same core, conserved machinery for different mitotic strategies.**⁶

>A corollary of our hypothesis is that closed mitosis has can persist... (line 299)

The “has” may have to be omitted.

Thank you for noticing this error. It has been corrected.

2. More direct evidence using genetic tools, even a transient one

The new imaging data using FM 4-64 has provided a piece of solid evidence that may compensate the lack of genetic manipulation. The readers would be convinced that *S. arctica* undergoes closed mitosis.

We appreciate your feedback, which helped us significantly strengthen the section of our manuscript describing the closed mitosis of *S. arctica*.

> Despite our efforts to replicate these finding in one of our laboratories...

Although what I have actually found is Suga & Miller (2018), where they disturbed the tyrosine kinase signalling using the transient overexpression, I have understood that the genetic manipulation technique in ichthyosporean is still too immature.

What I am concerned is that the readers may wonder why the authors used *S. arctica*, but not *C. fragrantissima* at all, on which there is an open (even though still narrow) path to directly address the question using the genetic tools. Please add a small clarification that the transformation efficiency is too low (or any other appropriate description) in *C. fragrantissimo* do so, perhaps at the sentence where the authors refer *S. arctica* as the best-studied ichthyosporean model.

This is a reasonable comment, and we have included the following text in the methods section as follows:

Line 449-452: The model ichthyosporean *S. arctica* with established protocols for cell synchronisation, live cell imaging and cytoskeletal inhibitor assays was selected as a representative Ichthyophonid for this study^{21,30,31}. Our attempts at replicating the reported genetic transformation of *C. fragrantissima*⁶⁰ were unsuccessful.

Reading through the revised manuscript, I have two more comments.

1. The word “cellularisation” is spelled “cellularization” at line 69.
2. The abbreviation “Cfra” is not necessary in Fig. 1.

We have corrected these errors.